# Rain or Snow: Hydrologic Processes, Observations, Prediction, and Research Needs

Adrian A. Harpold*, University of Nevada, Reno, aharpold@cabnr.unr.edu

Michael Kaplan, Desert Research Institute, michael.kaplan@dri.edu

P. Zion Klos, University of Idaho, zion.klos@gmail.com

Timothy Link, University of Idaho, tlink@uidaho.edu

James P. McNamara, Boise State University, jmcnamar@boisestate.edu

Seshadri Rajagopal, Desert Research Institute, Seshadri.Rajagopal@dri.edu

Rina Schumer, Desert Research Institute, rina@dri.edu

Caitriana M. Steele, New Mexico State University, caiti@nmsu.edu

*Corresponding author

**Abstract**
The phase of precipitation when it reaches the ground is a first-order driver of hydrologic
processes in a watershed. The presence of snow, rain, or mixed phase precipitation affects the
initial and boundary conditions that drive hydrological models. Despite their foundational
importance to terrestrial hydrology, typical phase prediction methods (PPM) specify phase based
on near-surface air temperature only. Our review conveys the diversity of tools available for
PPM in hydrological modeling and the advancements needed to improve predictions in complex
terrain with large spatiotemporal variations in precipitation phase.  Initially, we review the
processes and physics that control precipitation phase as relevant to hydrologists, focusing on the
importance of processes occurring aloft. There is a wide range of options for field observations
of precipitation phase, but there is a lack of a robust observation networks in complex terrain.
New remote sensing observations have potential to increase PPM fidelity, but generally require
assumptions typical of other PPM and field validation before they are operational. We review
common PPM and find that accuracy is generally increased at finer measurement intervals and
by including humidity information. One important tool for PPM development is atmospheric
modeling, which includes microphysical schemes that have not been effectively linked to
hydrological models or validated against near-surface precipitation phase observations. The
review concludes by describing key research gaps and recommendations to improve PPM,
including better incorporation of atmospheric information, improved validation datasets, and
regional-scale gridded data products. Two key points emerge from this synthesis for the
hydrologic community: 1) current PPM are too simple to capture important processes and are not
well-validated for most locations, 2) lack of sophisticated PPM increases the uncertainty in
estimation of hydrological sensitivity to changes in precipitation phase at local to regional scales.
The advancement of PPM is a critical research frontier in hydrology that requires scientific
cooperation between hydrological and atmospheric modelers and field scientists.
**Keywords**: precipitation phase, snow, rain, hydrological modeling
1.  Introduction and Motivation
As climate warms, a major hydrologic shift in precipitation phase from snow to rain is expected
to occur across temperate regions that are reliant on mountain snowpacks for water resource
provisioning (Bales et al., 2006; Barnett et al., 2005). Continued changes in precipitation phase
are expected to alter snowpack dynamics and both streamflow timing and amounts (Cayan et al.,
2001; Fritze et al., 2011; Luce and Holden, 2009; Klos et al., 2014; Berghuijs et al., 2014; Jepsen
et al., 2016), increase rain-on snow flooding (McCabe et al., 2007), and challenge our ability to
make accurate water supply forecasts (Milly et al., 2008). Accurate estimations of precipitation
inputs are required for effective hydrological modeling in both applied and research settings.
Snow storage delays the transfer of precipitation to surface runoff, infiltration, and generation of
streamflows (Figure 1), affecting the timing and magnitude of peak flows (Wang et al., 2016),
hydrograph recession (Yarnell et al., 2010) and the magnitude and duration of summer baseflow
(Safeeq et al., 2014; Godsey et al., 2014). Moreover, the altered timing and rate of snow versus
rain inputs can modify the partitioning of water to evapotranspiration versus runoff (Wang et al.,
2013). Misrepresentation of precipitation phase within hydrologic models thus propagates into
spring snowmelt dynamics (Harder and Pomeroy, 2013; Mizukami et al., 2013; White et al.,
2002; Wen et al., 2013) and streamflow estimates used in water resource forecasting (Figure 1).
The persistence of streamflow error is particularly problematic for hydrological models that are
calibrated on observed streamflows because this error can be compensated for by altering
parameters that control other states and fluxes in the model (Minder, 2010; Shamir and
Georgakakos, 2006; Kirchner, 2006). Expected changes in precipitation phase from climate
warming presents a new set of challenges for effective hydrological modeling (Figure 1). A
simple yet essential issue for nearly all runoff generation questions is this: Is precipitation falling
as rain, snow, or a mix of both phases?

Despite advances in terrestrial process-representation within hydrological models in the past
several decades (Fatichi et al., 2016), most state-of-the-art models rely on simple empirical
algorithms to predict precipitation phase. For example, nearly all operational models used by the
National Weather Service River Forecast Centers in the United States use some type of
temperature-based precipitation phase partitioning method (PPM) (Pagano et al., 2014). These
are often single or double temperature threshold models that do not consider other conditions
important to the hydrometeor's energy balance. Although forcing datasets for hydrological
models are rapidly being developed for a suite of meteorological variables, to date no gridded
precipitation phase product has been developed over regional to global scales. Widespread
advances in both simulation of terrestrial hydrological processes and computational capabilities
may have limited improvements on water resources forecasts without commensurate advances in
PPM.

Recent advances in PPM incorporate effects of humidity (Harder and Pomeroy, 2013; Marks et
al., 2013), atmospheric temperature profiles (Froidurot et al., 2014), and remote sensing of phase
in the atmosphere (Minder, 2010; Lundquist et al., 2008). A challenge to improving and selecting
PPM is the lack of validation data. In particular, reliable ground-based observations of phase are
sparse, collected at the point scale over limited areas, and are typically limited to research rather
than operational applications (Marks et al., 2013). The lack of observations is particularly
problematic in mountain regions where snow-rain transitions are widespread and critical for
regional water resource evaluations (Klos et al., 2014). For example, direct visual observations
have been widely used (Froidurot et al., 2014; Knowles et al., 2006; U.S. Army Corps of
Engineers, 1956), but are decreasing in number in favor of automated measurement systems.
Automated systems use indirect methods to accurately estimate precipitation phase from
hydrometeor characteristics (i.e. disdrometers), as well as coupled measurements that infer
precipitation phase based on multiple lines of evidence (e.g. co-located snow depth and
precipitation). Remote sensing is another indirect method that typically uses radar returns from
ground and space-borne platforms to infer hydrometeor temperature and phase. A comprehensive
description of the advantages and disadvantages of current measurement strategies, and their
correspondence with conventional PPM, is needed to determine critical knowledge gaps and
research opportunities.

New efforts are needed to advance PPM to better inform hydrological models by integrating new
observations, expanding the current observation networks, and testing techniques over regional
variations in hydroclimatology. While calls to integrate atmospheric information are an
important avenue for advancement (Feiccabrino et al., 2013), hydrological models ultimately
require accurate and validated phase determination at the land surface. Moreover, any
advancement that relies on integrating new information or developing a new PPM technique will
require validation and training using ground-based observations. To make tangible hydrological
modeling advancements, new techniques and datasets must be integrated with current modeling
tools. The first step towards improved hydrological modeling in areas with mixed precipitation
phase is educating the scientific community about current techniques and limitations that convey
the areas where research is most needed.
Our review paper is motivated by a lack of a comprehensive description of the state-of-the-art
PPM and observation tools. Therefore, we describe the current state of the science in a way that
clarifies the correspondence between techniques and observations, and highlights strengths and
weaknesses in the current scientific understanding. Specifically, subsequent sections will review:
1) the processes and physics that control precipitation phase as relevant to field hydrologists, 2)
current available options for observing precipitation phase and related measurements common in
remote field settings, 3) existing methods for predicting and modeling precipitation phase, and 4)
research gaps that exist regarding precipitation phase estimation. The overall objective is to
convey a clear understanding of the diversity of tools available for PPM in hydrological
modeling and the advancements needed to improve predictions in complex terrain characterized
by large spatiotemporal variations in precipitation phase.
2. Processes and Physics Controlling Precipitation Phase
Precipitation formed in the atmosphere is typically a solid in the mid-latitudes and its phase at
the land surface is determined by whether it melts during its fall (Stewart et al., 2015). Most
hydrologic models do not simulate atmospheric processes and specify precipitation phase based
on surface conditions alone (see Section 4.1), ignoring phase transformations in the atmosphere.
Several important properties that influence phase changes in the atmosphere are not included in
hydrological models (Feiccabrino et al., 2012), such as temperature and precipitation
characteristics (Theriault and Stewart, 2010), stability of the atmosphere (Theriault and Stewart,
2007), position of the 0 °C isotherm (Minder, 2010; Theriault and Stewart, 2010), interaction
between hydrometeors (Stewart, 1992), and the atmospheric humidity profile (Harder and
Pomeroy, 2013). The vertical temperature and humidity (represented by the mixing ratio) profile
through which the hydrometeor falls typically consists of three layers, a top layer that is frozen
(T <0 °C) in winter in temperate areas (Stewart, 1992), a mixed layer where T can exceed 0 °C,
and a surface layer that can be above or below 0 °C (Figure 2). The phase of precipitation at the
surface partly depends on the phase reaching the top of the surface layer, which is defined as the
critical height. The temperature profile and depth of the surface layer control the precipitation
phase reaching the ground surface. For example, in Figure 2a, if rain reaches the critical height, it
may reach the surface as rain or ice pellets depending on small differences in temperature in the
surface layer (Theriault and Stewart, 2010). Similarly, in Figure 2b, if snow reaches the critical
height, it may reach the surface as snow if the temperature in the surface layer is below freezing.
However, in Figure 2c, when the surface layer temperatures are close to freezing and the mixing
ratios are neither close to saturation nor very dry, the phase at the surface is not easily
determined by the surface conditions alone.

In addition to strong dependence on the vertical temperature and humidity profiles, precipitation
phase is also a function of fall rate and hydrometeor size because they affect energy exchange
with the atmosphere (Theriault et al., 2010). Precipitation rate influences the precipitation phase;
for example, a precipitation rate of 10 mm h$^{-1}$ reduces the amount of freezing rain by a factor of
three over a precipitation rate of 1 mm h$^{-1}$ (Theriault and Stewart, 2010) because there is less
time for turbulent heat exchange with the hydrometeor. A solid hydrometeor that originates in
the top layer and falls through the mixed layer can reach the surface layer as wet snow, sleet, or
rain. This phase transition in the mixed layer is primarily a function of latent heat exchange
driven by vapor pressure gradients and sensible heat exchange driven by temperature gradients.
Temperature generally increases from the mixed layer to the surface layer causing sensible heat
inputs to the hydrometeor. If these gains in sensible heat are combined with minimal latent heat
losses resulting from low vapor pressure deficits, it is likely that the hydrometeor will reach the
surface layer as rain (Figure 2). However, vapor pressure in the mixed layer is often below
saturation leading to latent energy losses and cooling of the hydrometeor coupled with diabatic
cooling of the local atmosphere, which can produce snow or other forms of frozen precipitation
at the surface even when temperatures are above 0 °C. Likewise, surface energetics affect local
atmospheric conditions and dynamics, especially in complex terrain. For example, melting of the
snowpack can cause diabatic cooling of the local atmosphere and affect the phase of
precipitation, especially when air temperatures are very close to 0 °C (Theriault et al., 2012).
Many conditions lead to a combination of latent heat losses and sensible heat gains by
hydrometeors (Figure 2). Under these conditions it can be difficult to predict the phase of
precipitation without sufficient information about humidity and temperature profiles, turbulence,
hydrometeor size, and precipitation intensity.

Stability of the atmosphere can also influence precipitation phase. Stability is a function of the
vertical temperature structure which can be altered by vertical air movement and hence influence
precipitation phase (Theriault and Stewart, 2007). Vertical air velocity changes the temperature
structure by adiabatic warming or cooling due to pressure changes of descending and ascending
air parcels, respectively. These changes in temperature can generate under-saturated or
supersaturated conditions in the atmosphere that can also alter the precipitation phase. Even a
very weak vertical air velocity (< 10 cm/s) significantly influences the phase and amount of
precipitation formed in the atmosphere (Theriault and Stewart, 2007). The rain-snow line
predicted by atmospheric models is very sensitive to these microphysics (Minder, 2010) and
validating the microphysics across locations with complex physiography is challenging.
Incorporation and validation of atmospheric microphysics is rarely achieved in hydrological
applications (Feiccabrino et al., 2015).

3. Current Tools for Observing Precipitation Phase
3.1 In situ observations
In situ observations refer to methods wherein a person or instrument onsite records precipitation
phase. We identify 3 classes of approaches that are used to observe precipitation phase including
1) direct observations, 2) coupled observations, and 3) proxy observations.

Direct observations simply involve a person on-site noting the phase of falling precipitation.
Such data form the basis of many of the predictive methods that are widely used (Dai, 2008;
Ding et al., 2014; U.S. Army Corps of Engineers, 1956). Direct observations are useful for
"manned" stations such as those operated by the U.S. National Weather Service. Few research
stations however, have this benefit, particularly in many remote regions and in complex terrain.
Direct observations are also limited in their temporal resolution and are typically reported only
once per day, with some exceptions (Froidurot et al., 2014). Citizen scientist networks have
historically provided valuable data to supplement primary instrumented observation networks.
The National Weather Service Cooperative Observer Program
(http://www.nws.noaa.gov/om/coop/what-is-coop.html, accessed 10/12/2016) is comprised of a

network of volunteers recording daily observations of temperature and precipitation, including phase. The NOAA National Severe Storms Laboratory used citizen scientist observations of rain and snow occurrence to evaluate the performance of the Multi-Radar Multi-Sensor (MRMS) system in the meteorological Phenomena Identification Near the Ground (mPING) project (Chen et al., 2015). The mPING project has recently been expanded to allow citizen scientists worldwide to easily report precipitation phase and characteristics using GPS-enabled smartphone applications (http://mping.nssl.noaa.gov, accessed 12/4/2016).  The Colorado Climate Center initiated the Community Collaborative Rain, Hail and Snow Network (CoCoRaHS) which supplies volunteers with low cost instrumentation to observe precipitation characteristics, including phase, and enables observations to be reported on the project website (http://www.cocorahs.org/, accessed 10/12/2016). Although highly valuable, some limitations of this system include the imperfect ability of observers to identify mixed phase events and the temporal extent of storms, as well as the lack of observations in both remote areas and during low light conditions.

Coupled observations link synchronous measurements of precipitation with secondary observations to indicate phase. Secondary observations can include photographs of surrounding terrain, snow depth measurements, and/or measurements of ancillary meteorological variables. Photographs of vertical scales emplaced in the snow have been used to estimate snow accumulation depth, which can then be coupled with precipitation mass to determine density and phase (Berris and Harr, 1987; Floyd and Weiler, 2008; Garvelmann et al., 2013; Hedrick and Marshall, 2014; Parajka et al., 2012). Mixed phase events however, are difficult to quantify using coupled depth- and photographic-based techniques (Floyd and Weiler, 2008). Acoustic distance sensors, which are now commonly used to monitor the accumulation of snow (e.g. Boe, 2013), have similar drawbacks in mixed phase events, but have been effectively applied to discriminate between snow and rain (Rajagopal and Harpold, 2016). Meteorological information such as temperature and relative humidity can be used to compute the phase of precipitation measured by bucket-type gauges. Unfortunately, this approach generally requires incorporating assumptions about the meteorological conditions that determine phase (see section 4.1). Harder and Pomeroy (2013) used a comprehensive approach to determine the phase of precipitation. Every 15 minutes during their study period phase was determined by evaluating weighing bucket mass, tipping

bucket depth, albedo, snow depth, and air temperature. Similarly, Marks et al. (2013) used a
scheme based on co-located precipitation and snow depth to discriminate phase. A more
involved expert decision making approach by L'hôte et al. (2005) was based on six recorded
meteorological parameters: precipitation intensity, albedo of the ground, air temperature, ground
surface temperature, reflected long-wave radiation, and soil heat flux. The intent of most of these
coupled observations was to develop datasets to evaluate PPM. However, if observation systems
such as these were sufficiently simple, they could have the potential to be applied operationally
across larger meteorological monitoring networks encompassing complex terrain where snow
comprises a large component of annual precipitation (Rajagopal and Harpold, 2016).

Proxy observations measure geophysical properties of precipitation to infer phase. The hot plate
precipitation gauge introduced by Rasmussen et al. (2012), for example, uses a thin heated disk
to accumulate precipitation and then measures the amount of energy required to melt snow or
evaporate liquid water. This technique however, requires a secondary measurement of air
temperature to determine if the energy is used to melt snow or only evaporate rain. Disdrometers
measure the size and velocity of hydrometeors. Although the most common application of
disdrometer data is to determine the drop size distribution (DSD) and other properties of rain, the
phase of hydrometeors can be inferred by relating velocity and size to density. Some disdrometer
technologies, which can be grouped into impact, imaging, and scattering approaches (Loffler-
Mang et al., 1999), are better suited for describing snow than others. Impact disdrometers, first
introduced by Joss and Waldvogel (1967), use an electromechanical sensor to convert the
momentum of a hydrometeor into an electric pulse. The amplitude of the pulse is a function of
drop diameter. Impact disdrometers have not been commonly used to measure solid precipitation
due to the different functional relationships between drop size and momentum for solid and
liquid precipitation. Imaging disdrometers use basic photographic principles to acquire images of
the distribution of particles (Borrmann and Jaenicke, 1993; Knollenberg, 1970). The 2D Video
Disdrometer (2DVD) described by Kruger and Krajewski (2002) records the shadows cast by
hydrometeors onto photodetectors as they pass through two sheets of light. The shape of the
shadows enables computation of particle size, and shadows are tracked through both light sheets
to determine velocity. Although initially designed to describe liquid precipitation, recent work
has shown that the 2DVD can be used to classify snowfall according to microphysical properties
of single hydrometeors (Bernauer et al., 2016). The 2DVD has been used to classify known rain
and snow events, but little work has been performed to distinguish between liquid and solid
precipitation. Scattering or optical disdrometers, measure the extinction of light passing between
a source and a sensor (Hauser et al., 1984; Loffler-Mang et al., 1999). Like the other types,
optical disdrometers were originally designed for rain, but have been periodically applied to
snow (Battaglia et al., 2010; Lempio et al., 2007). In a comparison study by Caracciolo et al.
(2006), the PARSIVEL optical disdrometer, originally described by Loffler-Mang et al. (1999)
did not perform well against a 2DVD because of problems related to the detection of slow fall
velocities for snow. It may be possible to use optical disdrometers to distinguish between rain,
sleet, and snow based on the existence of distinct shapes of the size spectra for each precipitation
type. More research on the relationship between air temperature and the size spectra produced by
the optical disdrometer is needed (Lempio et al., 2007). In summary, disdrometers of various
types are valuable tools for describing the properties of rain and snow, but require further testing
and development to distinguish between rain and snow, as well as mixed phase events.

3.2 Ground-based remote sensing observations
Ground-based remote sensing observations have been available for several decades to detect
precipitation phase using radar. Until recently, most ground-based radar stations were operated
as conventional Doppler systems that transmit and receive radio waves with single horizontal
polarization. Developments in dual polarization ground radar such as those that function as part
of the U.S. National Weather Service NEXRAD network (NOAA, 2016), have resulted in
systems that transmit radio signals with both horizontal and vertical polarizations. In general,
ground-based remote sensing observations, either single or dual-pol, remain underutilized for
detecting precipitation phase and are challenging to apply in complex terrain (Table 3).

Ground-based remote sensing of precipitation phase using single-polarized radar systems
depends on detecting the radar bright band. Radio waves transmitted by the radar system, are
scattered by hydrometeors in the atmosphere, with a certain proportion reflected back towards
the radar antenna. The magnitude of the measured reflectivity (Z) is related to the size and the
dielectric constant of falling hydrometeors (White et al., 2002). Ice particles aggregate as they
descend through the atmosphere and their dielectric constant increases, in turn increasing Z

measured by the radar, creating the bright band, a layer of enhanced reflectivity just below the elevation of the melting level (Lundquist et al., 2008). Therefore, bright band elevation can be used as a proxy for the "snow level", the bottom of the melting layer where falling snow transforms to rain (White et al., 2010;White et al., 2002).

Doppler vertical velocity (DVV) is another variable that can be estimated from single-polarized vertically profiling radar. DVV gives an estimate of the velocity of falling particles; as snowflakes melt and become liquid raindrops, the fall velocity of the hydrometeors increases. When combined with reflectivity profiles, DVV helps reduce false positive detection of the bright band, which may be caused by phenomena other than snow melting to rain (White et al., 2002). First, DVV and Z are combined to detect the elevation of the bottom of the bright band. The algorithm then searches for maximum Z above the bottom of the bright band and determines that to be the bright band elevation (White et al., 2002). However, a test of this algorithm on data from a winter storm over the Sierra Nevada found root mean square errors of 326 to 457 m compared to ground observations when the bright band elevation was assumed to represent the surface transition from snow to rain (Lundquist et al., 2008). Snow levels in mountainous areas however, may also be overestimated by radar profiler estimates if they are unable to resolve spatial variations close to mountain fronts, since snow levels have been noted to persistently drop on windward slopes (Minder and Kingsmill, 2013). Despite the potential errors, the elevation of maximum Z may be a useful proxy for snow level in hydrometeorological applications in mountainous watersheds because maximum Z will always occur below the freezing level (Lundquist et al., 2008; White et al., 2010)

Few published studies have explored the value of bright band-derived phase data for hydrologic modeling. Maurer and Mass (2006) compared the melting level from vertically pointing radar reflectivity against temperature-based methods to assess whether the radar approach could improve determination of precipitation phase at the ground level. In that study, the altitude of the top of the bright band was detected and applied across the study basin. Frozen precipitation was assumed to be falling in model pixels above the altitude of the melting level and liquid precipitation was assumed to be falling in pixels below the altitude of the melting layer (Maurer and Mass, 2006). Maurer and Mass (2006) found that incorporating radar-detected melting layer

altitude improved streamflow simulation results. A similar study that used bright band altitude to
classify pixels according to surface precipitation type was not as conclusive; bright band altitude
data did not improve hydrologic model simulation results over those based on a temperature
threshold (Mizukami et al., 2013). Also, the potential of the method is limited to the availability
of vertically pointing radar; in complex, mountainous terrain the ability to estimate melting level
becomes increasingly challenging with distance from the radar.

Dual-polarized radar systems generate more variables than traditional single-polarized systems.
These polarimetric variables include differential reflectivity, reflectivity difference, the
correlation coefficient, and specific differential phase. Polarimetric variables respond to
hydrometeor properties such as shape, size, orientation, phase state, and fall behavior and can be
used to assign hydrometeors to specific categories (Chandrasekar et al., 2013; Grazioli et al.,
2015), or to improve bright band detection (Giangrande et al., 2008).

Various hydrometeor classification algorithms have been applied to X-, C-, and S-band
wavelengths. Improvements in these algorithms over recent years have seen hydrometeor
classification become an operational meteorological product (see Grazioli et al., 2015 for an
overview). For example, the U.S. National Severe Storms Laboratory (NSSL) developed a fuzzy-
logic hydrometeor classification algorithm for warm-season convective weather (Park et al.,
2009) and this algorithm has also been tested for cold-season events (Elmore, 2011). Its skill was
tested against surface observations of precipitation type but the algorithm did not perform well in
classifying winter precipitation because it could not account for re-freezing of hydrometeors
below the melting level (Figure 2, Elmore, 2011). Unlike warm season convective precipitation,
the freezing level during a cold-season precipitation event can vary spatially. This phenomenon
has prompted the use of polarimetric variables to first detect the melting layer, and then classify
hydrometeors (Boodoo et al., 2010; Thompson et al., 2014). Although there has been some
success in developing two-stage cold-season hydrometeor classification algorithms, there is little
in the published literature that explores the potential contributions of these algorithms for
partitioning snow and rain for hydrological modeling.

3.3 Space-based remote sensing observations
Spaceborne remote sensing observations typically use passive or active microwave sensors to
determine precipitation phase (Table 3). Many of the previous passive microwave systems were
challenged by coarse resolutions and difficulties retrieving snowfall over snow-covered areas.
More recent active microwave systems are advantageous for detecting phase in terms of
accuracy and spatial resolution, but remain largely unverified. Table 3 provides and overview of
these space-based remote sensing technologies that are described in more detail below.
Passive microwave radiometers detect microwave radiation emitted by the Earth's surface or
atmosphere. Passive microwave remote sensing has the potential for discriminating between
rainfall and snowfall because microwave radiation emitted by the Earth's surface propagates
through all but the densest precipitating clouds, meaning that radiation at microwave
wavelengths directly interacts with hydrometeors within clouds (Olson et al., 1996; Ardanuy,
1989). However, the remote sensing of precipitation in microwave wavelengths and the
development of operational algorithms is dominated by research focused on rainfall (Arkin and
Ardanuy, 1989); by comparison, snowfall detection and observation has received less attention
(Noh et al., 2009; Kim et al., 2008). This is partly explained by examining the physical processes
within clouds that attenuate the microwave signal. Raindrops emit low levels of microwave
radiation increasing the level of radiance measured by the sensor; in contrast, ice hydrometeors
scatter microwave radiation, decreasing the radiance measured by a sensor (Kidd and Huffman,
2011). Land surfaces have a much higher emissivity than water surfaces, meaning that emission-
based detection of precipitation is challenging over land because the high microwave emissions
mask the emission signal from raindrops (Kidd, 1998; Kidd and Huffman, 2011). Thus,
scattering-based techniques using medium to high frequencies are used to detect precipitation
over land. Moreover, microwave observations at higher frequencies (> 89 GHz) have been
shown to discriminate between liquid and frozen hydrometeors (Wilheit et al., 1982).
Retrieving snowfall over land areas from spaceborne microwave sensors can be even more
challenging than for liquid precipitation because existing snow cover increases microwave
emission. Depression of the microwave signal caused by scattering from airborne ice particles
may be obscured by increased emission of microwave radiation from the snow covered land
surface. Kongoli et al. (2003) demonstrated an operational snowfall detection algorithm that
accounts for the problem of existing snow cover. This group used data from the Advanced
Microwave Sounding Unit-A (AMSU-A), a 15-channel atmospheric temperature sounder with a
single high frequency channel at 89 GHz), and AMSU-B, a 5-channel high frequency microwave
humidity sounder. Both sensors were mounted on the NOAA-16 and -17 polar-orbiting satellites.
While the algorithm worked well for warmer, opaque atmospheres, it was found to be too noisy
for colder, clear atmospheres. Additionally, some snowfall events occur under warmer conditions
than those that were the focus of the study (Kongoli et al., 2003). Kongoli et al. (2015) further
adapted their methodology for the Advanced Technology Microwave Sounder (ATMS - onboard
the polar-orbiting Suomi National Polar-orbiting Partnership satellite) a descendant of the
AMSU sounders. The latest algorithm assesses the probability of snowfall using the logistic
regression and the principal components of seven high frequency bands at 89 GHz and above. In
testing, the Kongoli et al. (2015) algorithm has shown skill in detecting snowfall both at variable
rates and when snowfall is lighter and occurs in colder conditions. An alternative algorithm by
Noh et al. (2009) used physically-based, radiative transfer modeling in an attempt to improve
snowfall retrieval over land. In this case, radiative transfer modeling was used to construct an *a*
*priori* database of observed snowfall profiles and corresponding brightness temperatures. The
radiative transfer procedure yields likely brightness temperatures from modeling how ice
particles scatter microwave radiation at different wavelengths. A Bayesian retrieval algorithm is
then used to estimate snowfall over land by comparing measured and modeled brightness
temperatures (Noh et al., 2009). The algorithm was tested during the early and late winter for
large snowfall events (e.g. 60 cm depth in 12 hours). Late winter retrievals indicated that the
algorithm overestimated snowfall over surfaces with significant snow accumulation.

While results have been promising, the spatial resolution at which ATMS and other passive
microwave data are acquired is very coarse (15.8 to 74.8 km at nadir), making passive
microwave approaches more applicable for regional to continental scales. Temporal resolution of
the data acquisition is another challenge. AMSU instruments are mounted on 8 satellites; the
related ATMS is mounted on a single satellite and planned for two additional satellites.
However, the satellites are polar-orbiting, not geostationary, so it is probable that a precipitation
event could occur outside the field of view of one of the instruments.

Spaceborne active microwave or radar sensors measure the backscattered signal from pulses of
microwave energy emitted by the sensor itself. Much like the ground based radar systems, the
propagated microwave signal interacts with liquid and solid particles in the atmosphere and the
degree to which the measured return signal is attenuated provides information on the
atmospheric constituents. The advantage offered by spaceborne radar sensors over passive
microwave is the capability to acquire more detailed sampling of the vertical profile of the
atmosphere (Kulie and Bennartz, 2009). The first spaceborne radar capable of observing
snowfall is the Cloud Profiling Radar (CPR) onboard CloudSat (2006 – present). The CPR
operates at 94 GHz with an along-track (or vertical) resolution of ~1.5 km. Retrieval of dry
snowfall rate from CPR measurements of reflectivity have been shown to correspond with
estimates of snowfall from ground-based radars at elevations of 2.6 and 3.6 km above mean sea
level (Matrosov et al., 2008). Estimates at lower elevations, especially those in the lowest 1 km,
are contaminated by ground clutter. Alternative approaches, combining CPR data with ancillary
data have been formulated to account for this challenge (Kulie and Bennartz, 2009; Liu, 2008).
Known relationships between CPR reflectivity data and the scattering properties of non-spherical
ice crystals are used to derive snowfall at a given elevation above mean sea level; below this
elevation a temperature threshold derived from surface data is used to discriminate between rain
and snow events. Liu (2008) used 2 °C as the snow/rain threshold, whereas Kulie and Bennartz
(2009) used 0 °C as the snow/rain threshold. Temperature thresholds have been the subject of
much research and debate for discriminating precipitation phase, as is further discussed in
section 4.1.

CloudSat is part of the A-train or afternoon constellation of satellites, which includes Aqua, with
the Moderate Resolution Imaging Spectrometer (MODIS) and the Cloud–Aerosol Lidar and
Infrared Pathfinder Satellite Observations (CALIPSO) spacecraft with cloud-profiling Lidar. The
sensors onboard A-train satellites provided the unique combination of data to create an
operational snow retrieval product. The CPR Level 2 snow profile product (2C-SNOW-
PROFILE) uses vertical profile data from the CPR, input from MODIS and the cloud profiling
radar, as well as weather forecast data to estimate near surface snowfall (Kulie et al., 2016;
Wood et al., 2013). The performance of 2C-SNOW-PROFILE was tested by Cao et al. (2014).
This group found the product worked well in detecting light snow but performed less
satisfactorily under conditions of moderate to heavy snow because of the non-stationary effects
of attenuation on the returned radar signal.

The launch of the Global Precipitation Mission (GPM) core observatory in February 2014 holds
promise for the future deployment of operational snow detection products. Building on the
success of the Tropical Rainfall Monitoring Mission (TRMM), the GPM core observatory
sensors include the Dual-frequency Precipitation Radar (DPR) and GPM Microwave Imager
(GMI). The GMI has two millimeter wave channels (166 and 183 GHz) that are specifically
designed to detect and retrieve light rain and snow precipitation. These are more advanced than
the sensors onboard the TRMM spacecraft and permit better quantification of the physical
properties of precipitating particles, particularly over land at middle to high latitudes (Hou et al.,
2014). Algorithms for the GPM mission are still under development, and are partly being driven
by data collected during the GPM Cold Season Experiment (GCPEx) (Skofronick-Jackson et al.,
2015). Using airborne sensors to simulate GPM and DPR measurements, one of the questions
that the GCPEx hoped to address concerned the potential capability of data from the DPR and
GMI to discriminate falling snow from rain or clear air (Skofronick-Jackson et al., 2015). The
initial results reported by the GCPEx study echo some of the challenges recognized for ground-
based single polarized radar detection of snowfall. The relationship between radar reflectivity
and snowfall is not unique. For the GPM mission, it will be necessary to include more variables
from dual frequency radar measurements, multiple frequency passive microwave measurements,
or a combination of radar and passive microwave measurements (Skofronick-Jackson et al.,

473    2015).


4. Current Tools for Predicting Precipitation Phase
4.1 Prediction Techniques from Ground-Based Observations
Discriminating between solid and liquid precipitation is often based on a near-surface air
temperature threshold (Martinec and Rango, 1986; U.S. Army Corps of Engineers, 1956; L'hôte
et al., 2005). Four prediction methods have been developed that use near-surface air temperature
for discriminating precipitation phase: 1) static threshold, 2) linear transition, 3) minimum and
maximum temperature, and 4) sigmoidal curve (Table 1). A static temperature threshold applies
a single temperature value, such as mean daily temperature, where all of the precipitation above
the threshold is rain, and all below the threshold is snow. Typically this threshold temperature is
near 0 °C (Lynch-Stieglitz, 1994; Motoyama, 1990), but was shown to be highly variable across
both space and time (Kienzle, 2008; Motoyama, 1990; Braun, 1984; Ye et al., 2013). For
example, Rajagopal and Harpold (2016) optimized a single temperature threshold at Snow
Telemetry (SNOTEL) sites across the western U.S. to show regional variability from -4 to 3 °C
(Figure 3). A second discrimination technique is to linearly scale the proportion of snow and rain
between a temperature for all rain ($T_{rain}$) and a temperature for all snow ($T_{snow}$) (Pipes and Quick,
1977; McCabe and Wolock, 2010; Tarboton et al., 1995). Linear threshold models have been
parameterized slightly differently across studies, e.g.: $T_{snow}$ =-1.0 °C, $T_{rain}$ = 3.0 °C (McCabe and
Wolock, 2010), $T_{snow}$ =-1.1 °C and $T_{rain}$ =3.3 °C (Tarboton et al., 1995), and $T_{snow}$ =0 °C and $T_{rain}$
=5 °C (McCabe and Wolock, 1999b). A third technique specifies a threshold temperature based
on daily minimum and maximum temperatures to classify rain and snow, respectively, with a
threshold temperature between the daily minimum and maximum producing a proportion of rain
and snow (Leavesley et al., 1996). This technique can have a time-varying temperature threshold
or include a $T_{rain}$ that is independent of daily maximum temperature. A fourth technique applies a
sigmoidal relationship between mean daily (or sub daily) temperature and the proportion or
probability of snow versus rain. For example, one method derived for southern Alberta, Canada
employs a curvilinear relationship defined by two variables, a mean daily temperature threshold
where 50% of precipitation is snow, and a temperature range where mixed-phase precipitation
can occur (Kienzle, 2008). Another sigmoidal-based empirical model identified a hyperbolic
tangent function defined by four parameters to estimate the conditional snow (or rain) frequency
based on a global analysis of precipitation phase observations from over 15,000 land-based
stations (Dai, 2008). Selection of temperature-based techniques is typically based on available
data, with a limited number of studies quantifying their relative accuracy for hydrological
applications (Harder and Pomeroy, 2014).

Several studies have compared the accuracy of temperature-based PPM to one another and/or
against an independent validation of precipitation phase. Sevruk (1984) found that only about
68% of the variability in monthly observed snow proportion in Switzerland could be explained
by threshold temperature based methods near 0 °C. An analysis of data from fifteen stations in
southern Alberta, Canada with an average of >30 years of direct observations noted over-
estimations in the mean annual snowfall for static threshold (8.1%), linear transition (8.2%),
minimum and maximum (9.6%), and sigmoidal transition (7.1%) based methods (Kienzle, 2008).
An evaluation of PPM at three sites in the Canadian Rockies by Harder and Pomeroy (2013)
found the largest percent error to occur using a static threshold (11% to 18%), followed by linear
relationships (-8% to 11%), followed by sigmoidal relationships (-3 to 11%). Another study
using 824 stations in China with >30 years of direct observations found accuracies of 51.4%
using a static 2.2 °C threshold and 35.7% to 47.4% using linear temperature-based thresholds
(Ding et al., 2014). Lastly, for multiple sites across the rain-snow transition in southwestern
Idaho, static temperature thresholds produced the lowest proportion (68%) whereas a linear-
based model produced the highest proportion (75%) of snow, respectively (Marks et al., 2013).
These accuracy assessments generally demonstrated that static threshold methods produced the
greatest errors, whereas sigmoidal relationships produced the smallest errors, although variations
to this general rule existed across sites.

Near surface humidity also influences precipitation phase (see Section 2). Three humidity-
dependent precipitation phase identification methods are found in the literature: 1) dewpoint
temperature ($T_d$), 2) wet bulb temperature ($T_w$), and 3) psychometric energy balance. The
dewpoint temperature is the temperature at which an air parcel with a fixed pressure and
moisture content would be saturated. In one approach to account for measurement and
instrument calibration uncertainties of ±0.25 °C each, $T_d$ and $T_w$ below -0.5 °C was assumed to
be all snow and above +0.5 °C all rain, with a linear relationship between the two being a
proportional mix of snow and rain (Marks et al., 2013). $T_d$ of 0.0 °C performed consistently
better than $T_a$ in one study by Marks et al. (2001) while a $T_d$ of 0.1°C for multiple stations in
Sweden was less accurate than a $T_a$ of 1.0 °C (Feiccabrino et al., 2013). The wet or ice bulb
temperature ($T_w$) is the temperature at which an air parcel would become saturated by
evaporative cooling in the absence of other sources of sensible heat, and is the lowest
temperature that falling precipitation can reach. Few studies have investigated the feasibility of
$T_w$ for precipitation phase prediction (Olsen, 2003; Ding et al., 2014; Marks et al., 2013). $T_w$
significantly improved prediction of precipitation phase over $T_a$ at 15-minute time steps, but only
marginally improved predictions at daily time steps (Marks et al., 2013). Ding et al. (2014)
developed a sigmoidal phase probability curve based on $T_w$ and elevation that outperformed $T_a$
threshold-based methods across a network of sites in China. Conceptually, the hydrometeor
temperature ($T_i$) is similar to $T_w$ but is calculated using the latent heat and vapor density gradient.
Use of computed $T_i$ values significantly improved precipitation phase estimates over $T_a$,
particularly as time scales approached one day (Harder and Pomeroy, 2013).

There has been limited validation of humidity-based precipitation phase prediction techniques
against ground-truth observations. Ding et al. (2014) showed that a method based on $T_w$ and
elevation increased accuracy by 4.8% to 8.9% over several temperature-based methods. Their
method was more accurate than a simpler $T_w$ based method by Yamazaki (2001). Feiccabrino et
al. (2013) showed that $T_d$ misclassified 3.0% of snow and rain (excluding mixed phase
precipitation), whereas $T_a$ only misclassified 2.4%. Ye et al. (2013) found $T_d$ less sensitive to
phase discrimination under diverse environmental conditions and seasons than $T_a$. Froidurot et
al. (2014) evaluated several techniques with a critical success index (CSI) at sites across
Switzerland to show the highest CSI values were associated with variables that included $T_w$ or
relative humidity (CSI=84%-85%) compared to $T_a$ (CSI=78%). Marks et al. (2013) evaluated the
time at which precipitation transitioned from snow to rain against field observations across a
range of elevations and found that $T_d$ most closely predicted the timing of phase change, whereas
both $T_a$ and $T_w$ estimated earlier phase changes than observed. Harder and Pomeroy (2013)
compared $T_i$ with field observations and found that error was <10% when $T_i$ was allowed to vary
with each daily time-step and >10% when $T_i$ was fixed at 0 °C. The $T_i$ accuracy increased
appreciably (i.e. 5%-10% improvement) when the temporal resolution was decreased from daily
to hourly or 15-minute time steps. The validation studies consistently showed improvements in
accuracy by including humidity over PPM based only on temperature.

Hydrological models employ a variety of techniques for phase prediction using ground-based
observations (Table 2). All discrete hydrological models (i.e. not coupled to an atmospheric
model) investigated used temperature based thresholds that did not consider the near-surface
humidity. Moreover, most models use a single static temperature threshold that typically
produces lower accuracy than multiple temperature methods. It should be noted that many of
these hydrological models lump by elevation zone, which improves estimates of the snow to rain
transition elevation and phase prediction accuracy in complex terrain compared to models
without elevation zones.  Hydrological models that are coupled to atmospheric models were
more able to consider important controls on precipitation phase, such as humidity and
atmospheric profiles. This compendium of model PPM highlights the current shortcomings in
phase prediction in conventional discrete hydrological models.
4.2 Prediction Techniques Incorporating Atmospheric Information
While many hydrologic models have their own formulations for determining precipitation phase
at the ground, it is also possible to initialize hydrologic models with precipitation phase fraction,
intensity, and volume from numerical weather simulation model output. Here we discuss the
limitations of precipitation phase simulation inherent to the Weather Research and Forecasting
(WRF) model (Kaplan et al., 2012; Skamarock et al., 2008) and other atmospheric simulation
models. The finest scale spatial resolution employed in atmospheric simulation models is ~1 km
and these models generate data at hourly or finer temporal resolutions. Regional climate models
(RCM) and global climate models (GCM) are typically coarser than local mesoscale models. The
physical processes driving both the removal of moisture from the air and the precipitation phase
(Section 2) occur at much finer spatial and temporal resolutions in the real atmosphere than
models typically resolve, i.e. <1 km. As with all numerical models, the representation of sub-grid
scale processes requires parameterization. At typical scales considered, characterization of mixed
phase processes within a condensing cloud depends on both cloud microphysics and kinematics
of the surrounding atmosphere. Replicating cloud physics at the multi-kilometer scale requires
empiricism. The 30+ cloud microphysics parameterization options in the research version of
WRF (Skamarock et al., 2008) vary in the number of classes described (cloud ice, cloud liquid,
snow, rain, graupel, hail, etc.), and may or may not accurately resolve changes in hydrometeor
phase and horizontal spatial location (due to wind) during precipitation. All microphysical
schemes predict cloud water and cloud ice based on internal cloud processes that include a
variety of empirical formulations or even simple lookup tables. These schemes vary greatly in
their accuracy with "mixed phase" schemes generally producing the most accurate simulations of
precipitation phase in complex terrain where much of the water is supercooled (Lin, 2007;
Reisner et al., 1998; Thompson et al., 2004; Thompson et al., 2008; Morrison et al., 2005; Zängl,
2007; Kaplan et al., 2012). Comprehensive validation of the microphysical schemes over
different land surface types (e. g. warm maritime, flat prairie, etc.) with a focus on different
snowfall patterns is lacking. In particular, in transition zones between mountains and plains or
along coastlines, the complexity of the microphysics becomes even more extreme due the
dynamics and interactions of differing air masses with distinct characteristics. The
autoconversion and growth processes from cloud water or ice to hydrometeors contain a strong
component of empiricism, and in particular, the nucleation media and their chemical
composition. Different microphysical parameterizations lead to different spatial distributions of
precipitation and produce varying vertical distributions of hydrometeors (Gilmore et al., 2004).
Regardless, precipitation rates for each grid cell are averages requiring hydrological modelers to
consider the effects of elevation, aspect, etc. in resolving precipitation phase fractions for finer-
scale models.

Numerical models that contain sophisticated cloud microphysics schemes allow assimilation of
additional remote sensing data beyond conventional synoptic/large scale observations (balloon
data). This is because the coarse spatial and temporal nature of radiosonde data results in the
atmosphere being sensed imperfectly/incompletely compared with the scale of motion that
weather simulation models can numerically resolve. These observational inadequacies are
exacerbated in complex terrain, where precipitation phase fraction can vary on small scales and
radar can be blocked by topography and therefore rendered useless in the model initialization.
Accurate generation of liquid and frozen precipitation from vapor requires accurate depiction of
initial atmospheric moisture conditions (Kalnay and Cai, 2003; Lewis et al., 2006). In
acknowledgement of the difficulty and uncertainty of initializing numerical simulation models,
atmospheric modelers use the term "bogusing" to describe incorporation of individual
observations at a point location into large scale initial conditions in an effort to enhance the
accuracy of the simulation (Eddington, 1989). They also employ complex assimilation
methodologies to force the early period of the model solutions during the time integration
towards fine scale observations (Kalnay and Cai, 2003; Lewis et al., 2006). These asynoptic or
fine scale data sources often substantially improve the accuracy of the simulations as time
progresses.

Hydrologists are increasingly using output from atmospheric models to drive hydrologic models
from daily to climatic or multi-decadal timescales (Tung and Haith, 1995; Pachauri, 2002; Wood
et al., 2004; Rojas et al., 2011; Yucel et al., 2015). These atmospheric models suffer from the
same data paucity and scale issues that likewise challenge the implementation and validation of
hydrologic models. Uncertainties in their output, including precipitation volume and phase,
begins with the initialization of the atmospheric model from measurements, increases with model
choice and microphysics as well as turbulence parameterizations, and is a strong function of the
scale of the model. The significance of these uncertainties varies by application, but should be
acknowledged. Furthermore, these uncertainties are highly variable in character and magnitude
from day to day and location to location. Thus, there has been very little published concerning
how well atmospheric models predict precipitation phase. Finally, lack of ground measurements
leaves hydrologists with no means to assess and validate atmospheric model predictions.

5. Research Gaps
The incorrect prediction of precipitation phase leads to cascading effects on hydrological
simulations (Figure 1). Meeting the challenge of accurately predicting precipitation phase
requires the closing of several critical research gaps (Figure 4). Perhaps the most pressing
challenge for improving PPM is developing and employing new and improved sources of data.
However, new data sources will not yield much benefit without effective incorporation into
predictive models (Figure 4). Additionally, both the scientific and management communities
lack data products that can be readily understood and broadly used. Addressing these research
gaps requires simultaneous engagement both within and between the hydrology and atmospheric
observation and modeling communities. Changes to atmospheric temperature and humidity
profiles from regional climate change will likely challenge conventional precipitation phase
prediction in ways that demand additional observations and improved forecasts.

We also highlight research gaps to improve relatively simple hydrological models without
adding unnecessary complexity associated with sophisticated PPM approaches.  For example,
more efforts to verify the existing PPM in different climatic environments and during specific
hydrometeorological events could help determine various temperature thresholds (Table 1) to
apply in existing models (section 5.3).  In addition, developing gridded precipitation phase
products may eliminate the need to make existing models more complex by applying more
complex PPM outside of those models, e.g. similar to precipitation distribution in existing
gridded products used by many hydrological models. Ultimately, recognizing the sensitivity of
hydrological model outcomes to PPM and identifying what climates and applications require
higher phase prediction accuracy are crucial steps to determining the complexity of PPM
required for specific applications.

5.1 Conduct focused field campaigns
Intensive field campaigns are extremely effective approaches to address fundamental research
gaps focused on the discrimination between rain, snow, and mixed-phase precipitation at the
ground by providing opportunities to test novel sensors, collect detailed datasets to develop
remote sensing retrieval algorithms and improve PPM estimation methods. The recent Global
Precipitation Measurement (GPM) Cold Season Precipitation Experiment (GCPEx) is an
example of such a campaign in non-complex terrain where simultaneous observations using
arrays of both airborne and ground-based sensors were used to measure and characterize both
solid and liquid precipitation (e.g. Skofronick-Jackson et al., 2015). Similar intensive field
campaigns are needed in complex terrain that is frequently characterized by highly dynamic and
spatially variable hydrometeorological conditions. Such campaigns are expensive to conduct, but
can be implemented as part of operational nowcasting to develop rich data resources to advance
scientific understanding as was very effectively done during the Vancouver Olympic Games in
2010 (Isaac et al., 2014; Joe et al., 2014). The research community should utilize existing
datasets and capitalize on similar opportunities and expand environmental monitoring networks
to simultaneously advance both atmospheric and hydrological understanding, especially in
complex terrain spanning the rain-snow transition zone.

5.2 Incorporate humidity information
Atmospheric humidity affects the energy budget of falling hydrometeors (Section 4.1), but is
rarely considered in precipitation phase prediction. The difficulty in incorporating humidity
mainly arises from a lack of observations, both as point measurements and distributed gridded
products. For example, while some reanalysis products have humidity information (i.e. National
Centers for Environmental Prediction, NCEP reanalysis) they are at spatial scales (i.e. > 1
degree) that are too coarse for resolving precipitation phase in complex topography. Addition of
high-quality aspirated humidity sensors at snow monitoring stations, such as the SNOTEL
network, would advance our understanding of humidity and its effects on precipitation phase in
the mountains. Because dry air masses have regional variations controlled by storm tracks and
proximity to water bodies, sensitivity of precipitation phase to humidity variations driven by
regional warming remains relatively unexplored.

Although humidity datasets are relatively rare in mountain environments, some gridded data
products exist that can be used to investigate the importance of humidity information. Most
interpolated gridded data products either do not include any measure of humidity (e.g. Daymet or
WorldClim) or use daily temperature measurements to infer humidity conditions (e.g. PRISM).
In complex terrain, air temperature can also vary dramatically at relatively small scales from
ridgetops to valley bottoms due to cold air drainage (Whiteman et al., 1999) and hence can
introduce errors into inferential techniques such as these. Potentially more useful are data
assimilation products, such as NLDAS-2, that provide humidity and temperature values at $1/8^{th}$
of a degree scale over the continental U.S. In addition, several data reanalysis products are often
available at 1 to 3 year lags from present, including NCEP/NCAR, NARR, and the $20^{th}$ Century
reanalysis. Given the relatively sparse observations of humidity in mountain environments, the
accuracy of gridded humidity products is rarely rigorously evaluated (Abatzoglou, 2013). More
work is needed to understand the added skill provided by humidity datasets for predicting
precipitation phase and its distribution over time and space.

5.2 Incorporate atmospheric information
We echo the call of Feiccabrino et al. (2015) for greater incorporation of atmospheric
information into phase prediction and additional verification of the skill in phase prediction
provided by atmospheric information.

Several avenues exist to better incorporate atmospheric information into precipitation phase
prediction, including direct observations, remote sensing observations, and synthetic products.
Radiosonde measurements made daily at many airports and weather forecasting centers have
shown some promise for supplying atmospheric profiles of temperature and humidity (Froidurot
et al., 2014). However, these data are only useful to initialize the larger scale structure of
temperature and water vapor, and may not capture local-scale variations in complex terrain. It is
also their lack of temporal and spatial frequency that prevents their use in accurate precipitation
phase prediction, which is inherently a mesoscale problem, i.e., scales of motion <100 km.
Atmospheric information on the bright-band height from Doppler radar has been utilized for
predicting the altitude of the rain-snow transition (Lundquist et al., 2008; Minder, 2010), but has
rarely been incorporated into hydrological modeling applications (Maurer and Mass, 2006;
Mizukami et al., 2013). In addition to atmospheric observations, modeling products that
assimilate observations or are fully physically-based may provide additional information for
precipitation phase prediction. Numerous reanalysis products (described in Section 2.2) provide
temperature and humidity at different pressure levels within the atmosphere. To our knowledge,
information from reanalysis products has yet to be incorporated into precipitation phase
prediction for hydrological applications. Bulk microphysical schemes used by meteorological
models (e.g. WRF) provide physically-based estimates of precipitation phase. These schemes
capture a wide-variety of processes, including evaporation, sublimation, condensation, and
aggradation, and output between two and ten precipitation types. Historically, meteorological
models have not been run at spatial scales capable of resolving convective dynamics (e.g. <2
km), which can exacerbate error in precipitation phase prediction in complex terrain with a moist
neutral atmosphere. Coarse meteorological models also struggle to produce pockets of frozen
precipitation from advection of moisture plumes between mountain ranges and cold air wedged
between topographic barriers. However, reduced computational restrictions on running these
models at finer spatial scales and over large geographic extents (Rasmussen et al., 2012) are
enabling further investigations into precipitation phase change under historical and future climate
scenarios. This suggests that finer dynamical downscaling is necessary to resolve precipitation
phase which is consistent with similar work attempting to resolve winter precipitation amount in
complex terrain (Gutmann et al., 2012). A potentially impactful area of research is to integrate
this information into novel approaches to improve precipitation phase prediction skill.

5.3 Disdrometer networks operating at high temporal resolutions
An increase in the types and reliability of disdrometers over the last decade has provided a new
suite of tools to more directly measure precipitation phase. Despite this new potential resource
for distinguishing snow and rain, very limited deployments of disdrometers have occurred at the
scale necessary to improve hydrologic modeling and rain-snow elevation estimates. The lack of
disdrometer deployment likely arises from a number of potential limitations: 1) known issues
with accuracy, 2) cost of these systems, and 3) power requirements needed for heating elements.
These limitations are clearly a factor in procuring large networks and deploying disdrometers in
complex terrain that is remote and frequently difficult to access. However, we advise that
disdrometers offer numerous benefits that cannot be substituted with other measurements: 1)
they operate at fine temporal scales, 2) they operate in low light conditions that limit other direct
observations, and 3) they provide land surface observations rather than precipitation phase in the
atmosphere (as compared to more remote methods). Moreover, improvements in disdrometer and
power supply technologies that address these limitations would remove restrictions on increased
disdrometer deployment.

Transects of disdrometers spanning the rain-snow elevations of key mountain areas could add
substantially to both prediction of precipitation phase for modeling purposes, as well as
validating typical predictive models. We advocate for transects over key mountain passes where
power is generally available and weather forecasts for travel are particularly important. In
addition, co-locating disdrometers at long-term research stations where precipitation phase
observations could be tied to micro-meteorological and hydrological observations has distinct
advantages. These areas often have power supplies and instrumentation expertise to operate and
maintain disdrometer networks.

5.4 Compare different indirect phase measurement methods
There is an important need to evaluate the accuracy of different PPM to assess tradeoffs between
model complexity and skill (Figure 4). Given the potential for several types of observations to
improve precipitation phase prediction (section 5.1-5.3), quantifying the relative skill provided
by these different lines of evidence is a critical research gap. Although assessing relative
differences between methods is potentially informative, comparison to ground truth
measurements is critical for assessing accuracy. Disdrometer measurements and video imaging
(Newman et al., 2009) are ideal ground truthing methods that can be employed at fine time steps
and under a variety of conditions (section 5.3). Less ideal for accuracy assessment studies are
direct visual observations that are harder to collect at fine time steps and in low light conditions.
Similarly, employing coupled observations of precipitation and snow depth has been used to
assess accuracy of different precipitation phase prediction methods (Marks et al., 2013; Harder
and Pomeroy, 2013), but accuracy assessment of these techniques themselves are lacking under a
wide range of contrasting hydrometeorological conditions.

A variety of accuracy assessments are needed that will require co-located distributed
measurements. One critical accuracy assessment involves the consistency of different
precipitation phase prediction methods under different climate and atmospheric conditions.
Assessing the effects of climate and atmospheric conditions requires measurements from a
variety of sites covering a range of hydroclimatic conditions and record lengths that span the
conceivable range of atmospheric conditions at a given site. Another important evaluation metric
is the performance over different time steps. Harder and Pomeroy (2013) showed that
hydrometeor and temperature-based prediction methods had errors that substantially decreased
across shorter time steps. Identifying the effects of time step length on the accuracy of different
prediction methods has been relatively unexplored, but is critical to select the most appropriate
method for specific hydrological applications. Finally, the performance metrics used to assess
accuracy should be carefully considered. The applications of precipitation phase prediction
methods are diverse, necessitating a wide variety of performance metrics, including the
probability of snow versus rain (Dai, 2008), the error in annual or total snow/rain accumulation
(Rajagopal and Harpold, 2016), performance under extreme conditions of precipitation amount
and intensity, determination of the snow-rain elevation (Marks et al., 2013), and uncertainty
arising from measurement error and accuracy. Comparison of different metrics across a wide-
variety of sites and conditions is lacking but is greatly needed to advance hydrologic science in
cold regions.

5.5 Develop spatially resolved products
Many hydrological applications will benefit from gridded data products that are easily integrated
into standard hydrological models. Currently, very few options exist for gridded data
precipitation phase products. Instead, most hydrological models have some type of submodel or
simple scheme that specifies precipitation phase as rain, snow, or mixed-phase precipitation (see
Section 4). While testing PPM with ground based observations could lead to improved
submodels, we believe development of gridded forcing data may be an easier and more effective
solution for many hydrological modeling applications.

Gridded data products could be derived from a combination of remote sensing and existing
synthetic products, but would need to be extensively evaluated. The NASA GPM mission is
beginning to produce gridded precipitation phase products at 3-hour and 0.1 degree resolution.
However, GPM phase is measured at the top of the atmosphere, typically relies on simple
temperature-thresholds, and has yet to be validated with ground based observations. Another
existing product is the Snow Data Assimilation System (SNODAS) that estimates liquid and
solid precipitation at the 1 km scale. However, the developers of SNODAS caution that it is not
suitable for estimating storm totals or regional differences. Furthermore, to our knowledge the
precipitation phase product from SNODAS has not been validated with ground observations. We
suggest the development of new gridded data products that utilize new PPM (i.e. Harder and
Pomeroy, 2013) and new and expanded observational datasets, such as atmospheric information
and radar estimates. We advocate for the development of multiple gridded products that can be
evaluated with surface observations to compare and contrast their strengths. Accurate gridded
phase products rely on the ability to represent the physics of water vapor and energy flows in
complex terrain (e.g. Holden et al., 2010) where statistical downscaling methods are typically
insufficient (Gutmann et al., 2012). This would also allow for ensembles of phase estimates to be
used in hydrological models, similar to what is currently being done with gridded precipitation
estimates.

5.6 Characterization of regional variability and response to climate change
The inclusion of new datasets, better validation of PPM, and development of gridded data
products will poise the hydrologic community to improve hydrological predictions and better
quantify regional sensitivity of phase change to climate changes. Because broad-scale techniques
applied to assess changes in precipitation phase and snowfall have relied on temperature, both
regionally (Klos et al., 2014; Pierce and Cayan, 2013; Knowles et al., 2006) and globally
(Kapnick and Delworth, 2013; O'Gorman, 2014), they have not fully considered the potential
non-linearities created by the absence of wet bulb depressions and humidity in assessment of
sensitivity to changes in phase. Consequently, the effects of changes from snow to rain from
warming and corresponding changes in humidity will be difficult to predict with current PPM.
Recent efforts by Rajagopal and Harpold (2016) have demonstrated that simple temperature
thresholds are insufficient to characterize snow-rain transition across the western U.S. (Figure 3),
perhaps because of differences in humidity. An increased focus on future humidity trends,
patterns, GCM simulation errors (Pierce et al., 2013) and availability of downscaled humidity
products at increasingly finer scales (e.g.: Abatzoglou, 2013; Pierce and Cayan, 2016) will
enable detailed assessments of the relative role of temperature and humidity in future
precipitation phase changes. Recent remote sensing platforms, such as GPM, may offer an
additional tool to assess regional variability, however, the current GPM precipitation phase
product relies on wet bulb temperatures based on model output and not microwave-based
observations (Huffman et al., 2015). In addition to issues with either spatial or temporal
resolution or coverage, one of the main challenges in using remotely sensed data for
distinguishing between frozen and liquid hydrometeors is the lack of validation. Where products
have been validated, the results are usually only relevant for the locale of the study area.
Spaceborne radar combined with ground-based radar offers perhaps the most promising solution,
but given the non-unique relationship between radar reflectivity and snowfall, further testing is
necessary in order to develop reliable algorithms.

Future work is needed to improve projections of changes in snowpack and water availability
from regional to global scales. This local to sub-regional characterization is needed for water
resource prediction and to better inform decision and policy makers. In particular, the ability to
predict the transitional rain-snow elevations and its uncertainty is critical for a variety of end-
users, including state and municipal water agencies, flood forecasters, agricultural water boards,
transportation agencies, and wildlife, forest, and land managers. Fundamental advancements in
characterizing regional variability are possible by addressing the research challenges detailed in
sections 5.1-5.5.

6. Conclusions
This review paper is a step towards communicating the potential bottlenecks in hydrological
modeling caused by poor representation of precipitation phase (Figure 1). Our goals are to
demonstrate that major research gaps in our ability to PPM are contributing to errors and
reducing the predictive skill of hydrological models. By highlighting the research gaps that could
advance the science of PPM, we provide a roadmap for future advances (Figure 4). While many
of the research gaps are recognized by the community and are being pursued, including
incorporating atmospheric and humidity information, others remain essentially unexplored (e.g.
production of gridded data, widespread ground validation, and remote sensing validation).

The key points that must be communicated to the hydrologic community and its funding
agencies can be distilled into the following two statements: 1) current PPM are too simple to
capture important processes and are not well-validated for most locations, 2) the lack of
sophisticated PPM increases the uncertainty in estimation of hydrological sensitivity to changes
in precipitation phase at local to regional scales. We advocate for better incorporation of new
information (5.1-5.2) and improved validation methods (5.3-5.4) to advance our current PPM
and observations. These improved PPM and remote-sensing observations will be capable of
developing gridded datasets (5.5) and providing new insight that reduce the uncertainty of
predicting regional changes from snow to rain (5.6).  Improved PPM and existing phase products
will also facilitate improvement of simpler hydrological models for which more complex PPM
are not justified.  A concerted effort by the hydrological and atmospheric science communities to
address the PPM challenge will remedy current limitations in hydrological modeling of
precipitation phase, advance of understanding of cold regions hydrology, and provide better
information to decision makers.

Acknowledgements
This work was conducted as a part of an Innovation Working Group supported by the Idaho,
Nevada, and New Mexico EPSCoR Programs and by the National Science Foundation under
award numbers IIA-1329469, IIA-1329470 and IIA-1329513. Adrian Harpold was partially
supported by USDA NIFA NEV05293. Adrian Harpold and Rina Schumer were supported by
the NASA EPSCOR Cooperative Agreement #NNX14AN24A. Timothy Link was partially
supported by the Department of the Interior Northwest Climate Science Center (NW CSC)
through a Cooperative Agreement #G14AP00153 from the United States Geological Survey
(USGS). Seshadri Rajagopal was partially supported by research supported by NSF/USDA grant
(#1360506/#1360507) and startup funds provided by Desert Research Institute. The contents of
this manuscript are solely the responsibility of the authors and do not necessarily represent the
views of the NW CSC or the USGS. This manuscript is submitted for publication with the
understanding that the United States Government is authorized to reproduce and distribute
reprints for Governmental purposes.

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

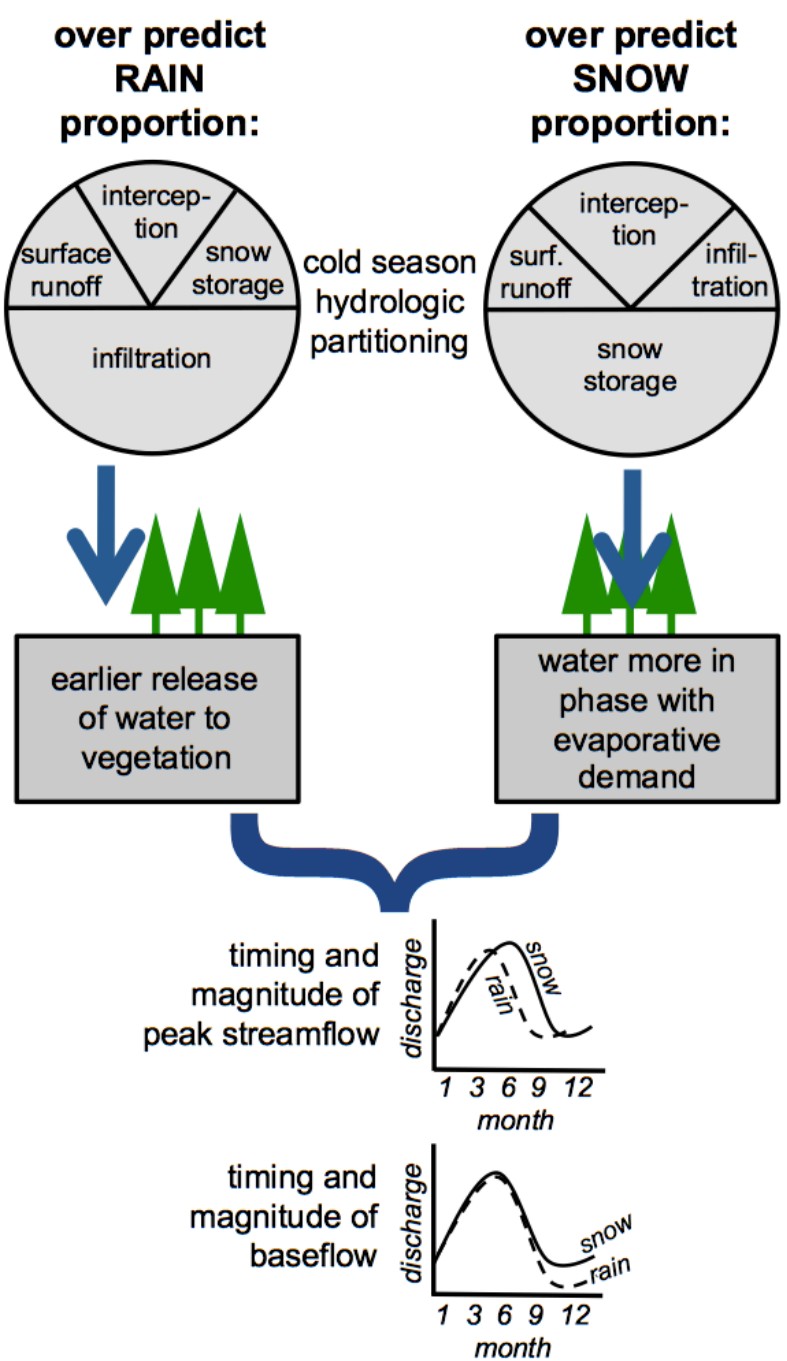


Figure 1: Precipitation phase has numerous implications for modeling the magnitude, storage,
partitioning, and timing of water inputs and outputs. Potentially affecting important
ecohydrological and streamflow quantities important for prediction.

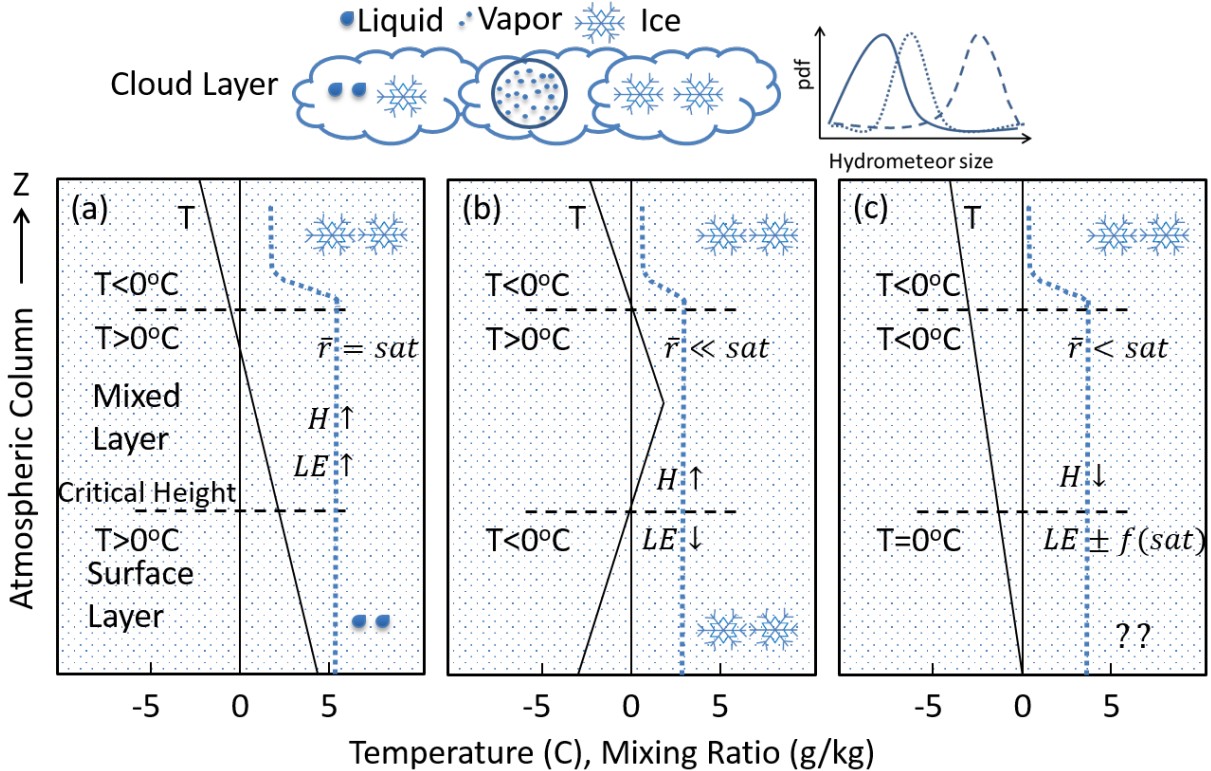


Figure 2: The phase of precipitation at the ground surface is strongly controlled by atmospheric
profiles of temperature and humidity. While conditions exist that are relatively easy to predict
rain (a) and snow (b), many conditions lead to complex heat exchanges that are difficult to
predict with ground based observations alone (c). The blue dotted line represents the mixing
ratio. H, LE, f(sat), and r are abbreviations for sensible heat, latent heat of evaporation, function
of saturation and mixing ratio respectively. The arrows after H or LE indicate the energy of the
hydrometeor either increasing (up) or decreasing (down) which is controlled by other
atmospheric conditions.


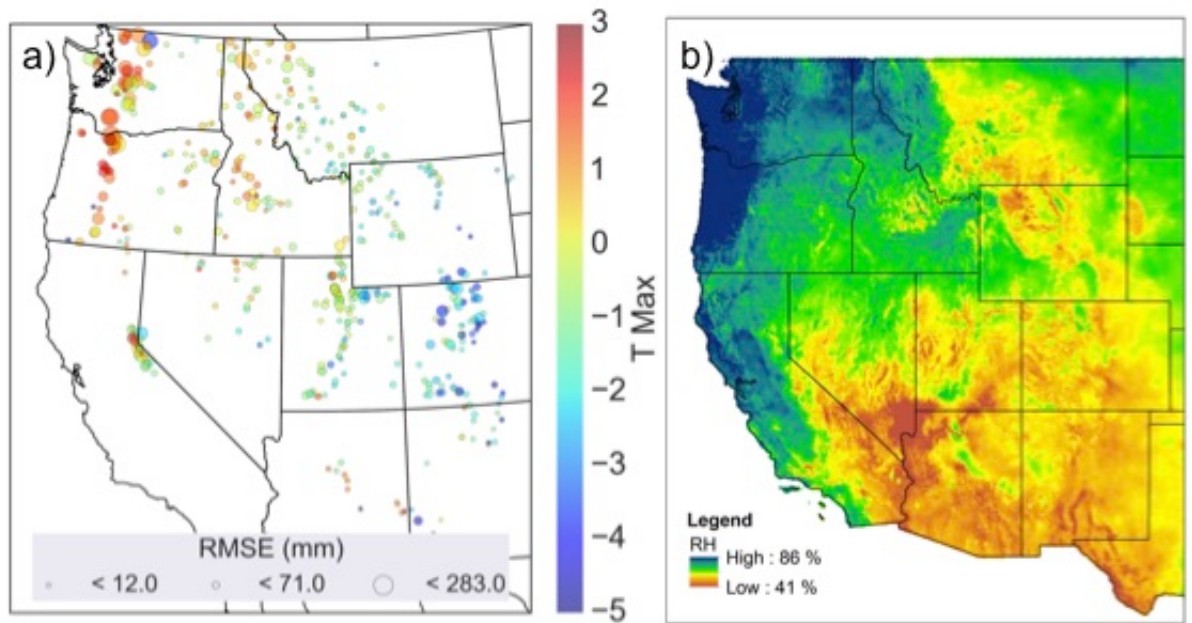


Figure 3: The optimized critical maximum daily temperature threshold that produced the lowest
Root Mean Square Error (RMSE) in the prediction of snowfall at Snow Telemetry (SNOTEL)
stations across the western US (adapted from Rajagopal and Harpold, 2016). b) Precipitation day
relative humidity averaged over 1981-2015 based on the Gridmet dataset (Abatzoglou, 2013).


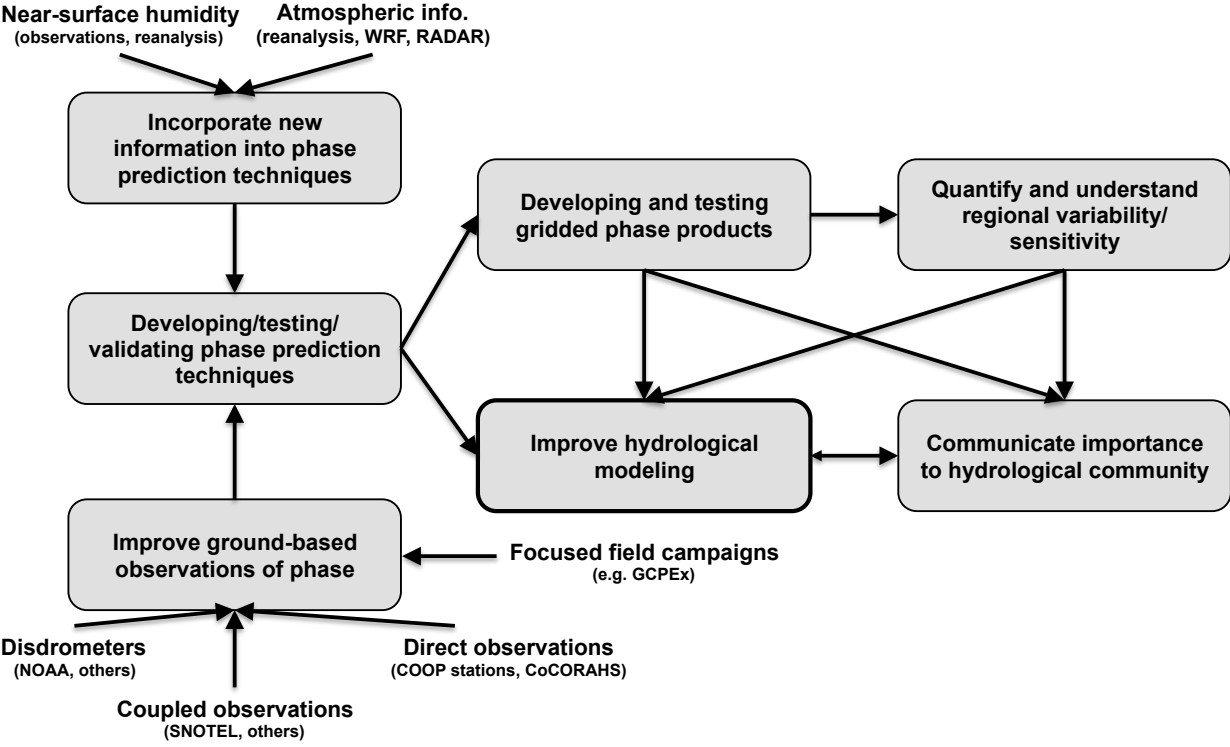


Figure 4: Conceptual representation of the research gaps and workflows needed to advance PPM
and improve hydrological modeling.

Table 1. Mathematical expression for the four common temperature-based PPM to estimate snow
fraction (S) or snow frequency (F) using the mean air temperature ($T_a$), max daily air
temperature ($T_{a\text{-}max}$), and/or minimum daily air temperature ($T_{a\text{-}min}$). The variable $T_{snow}$ is air
temperature when all precipitation (P) is snow and $T_{rain}$ is the air temperature when all air
precipitation is rain.

| Type | Mathematical expression for snow fraction (S) or snow frequency (F) | Reference(s) |
|------|---------------------------------------------------------------------|--------------|
| Static threshold | $$S = \begin{cases} P \ for \ T_a \leq T_{snow} \\ 0 \ for \ T_a \geq T_{snow} \end{cases}$$ | Motoyama, 1990 |
| Linear transition | $$S = \begin{cases} P \ for \ T_a \leq T_{snow} \\ P\left(\dfrac{T_{rain} - T_a}{T_{rain} - T_{snow}}\right) \ for \ T_{snow} < T_a < T_{rain} \\ 0 \ for \ T_a \geq T_{snow} \end{cases}$$ | McCabe and Wolock, 1998b |
| Minimum and maximum temperature | $$S = \begin{cases} P \ for \ T_{a-max} \leq T_{snow} \\ 1 - P\left[\dfrac{(T_{a-max} - T_{snow})}{(T_{a-max} - T_{a-min})}\right] \ for \ T_{snow} < T_{a-max} < T_{rain} \\ 0 \ for \ T_{a-max} \geq T_{rain} \end{cases}$$ | Leavesley, 1996 |
| Sigmoidal curve | $$S = P * a\left[\tanh\big(b(T_a - c)\big) - d\right]$$ $$F = a\left[\tanh\big(b(T_a - c)\big) - d\right]$$ | Dai, 2008 |





Table 2. Common hydrological models and the precipitation phase prediction (PPM) technique
employed. The citation referring to the original publication of the model is given.

| Model | PPM technique | Citations |
|---|---|---|
| Discrete Models (not coupled) | | |
| HBV | Static Threshold | Bergström, 1995 |
| Snowmelt Runoff Model | Static Threshold | Martinec et al., 2008 |
| SLURP | Static Threshold | Kite, 1995 |
| UBC Watershed Model | Linear Transition | Pipes and Quick, 1977 |
| PRMS model | Minimum & Maximum Temperature | Leavesley et al., 1996 |
| USGS water budget | Linear transition between two mean temps | McCabe and Wolock, 1999a |
| SAC-SMA (SNOW-17) | Static Threshold | Anderson, 2006 |
| DHSVM | Linear transition (double check) | Wigmosta et al., 1994 |
| SWAT | Threshold Model | Arnold et al., 2012 |
| RHESSys | Linear transition or input phase | Tague and Band, 2004 |
| HSPF | Air and dew point temperature thresholds | Bicknell et al., 1997 |
| THE ARNO MODEL | Static Threshold | Todini, 1996 |
| HEC-1 | Static Threshold | HEC-1, 1998 |
| MIKE SHE | Static Threshold | MIKE-SHE User Manual |
| SWAP | Static Threshold | Gusev and Nasonova, 1998 |
| BATS | Static Threshold | Yang et al., 1997 |
| Utah Energy Balance | Linear Transition | Tarboton and Luce, 1996 |
| SNOBAL/ISNOBAL | Linear Transition[*] | Marks et al., 2013 |
| CRHM | Static Threshold | Fang et al., 2013 |
| GEOTOP | Linear Transition | Zanotti et al. 2004 |
| SNTHERM | Linear Transition | SNTHERM Online Documentation |
| Offline LS models | | |
| Noah | Static Threshold | Mitchell et al., 2005 |
| VIC | Static Threshold | VIC Documentation |
| CLASS | Multiple Methods[+] | Verseghy, 2009 |


* by default. Temperature-phase-density relationship explicitly specified by user.
+ A flag is specified which switches between, static threshold, linear transition.







Table 3: Remote sensing technologies useful to precipitation phase discrimination organized into
ground-based, spaceborne with passive microwave, and passive with active microwave. The
table describes the variables of interest, their temporal and spatial coverage, and associated
references.

| Technology | Variables | Spatial resolution; coverage | Temporal resolution, period of record | References |
|---|---|---|---|---|
| **Ground-based systems** | | | | |
| Vertically pointing, single polarized 915-MHz Doppler wind profilers | Reflectivity, brightband height, Doppler vertical velocity | 100 m vertical resolution; deployed locally in Sierra Nevada basins | Hourly, Winters 1998, 2001 - 2005 | White et al., 2002; Lundquist et al., 2008 |
| NEXRAD Dual polarized radar | Reflectivity[1], hydrometeor classification[1], melting layer[1], hybrid hydrometeor classification[1] | 0.5° azimuthal by 250 m; range 460 km; Nationwide[2] | 5 - 10 minutes; 2011[3] - present | Giangrande et al., 2008; Park et al., 2009; Elmore, 2011; Grazioli et al., 2015 |
| **Spaceborne systems: Passive microwave** | | | | |
| NOAA-15, NOAA-16, NOAA-17 Advanced Microwave Sounding Unit-A, B | Brightness temperature | 48 km (AMSU-A), 16 km (AMSU-B); global coverage, with 22000 km swath | For two platforms, 6 hours revisit time; three platforms, 4 hours revisit time[4]; 1998 - present | Kongoli et al., 2003 |
| SUOMI-NPP Advanced Technology Microwave Sounder | Brightness temperature | 15 - 50 km; global coverage, with 2200 km swath | Daily; 2011 - present | Kongoli et al., 2015 |
| GPM Core Observatory Microwave Imager | Brightness temperature | 4.4 km by 7.3 km; global coverage, 904 km swath | 2014 to present | Skofronick-Jackson et al., 2015 |
| **Spaceborne systems: Active microwave** | | | | |
| Cloud Profiling Radar (CPR) | Radar reflectivity, 2C-SNOW-PROFILE | 1.4 by 1.7 km; swath 1.4 km | 16 days; 2006 to present | Wood et al., 2013; Cao et al., 2014; Kulie et al., 2016; |
| GPM Core Observatory Dual-frequency Precipitation Radar | Radar reflectivity | 5 km; global coverage, 120 - 245 km swath | 2 – 4 hours; 2014 to present | Skofronick-Jackson et al., 2015 |

*Notes:*
1. Operational products available from NOAA (2016). The operational products are not ground validated, except
where analyzed for specific studies.
2. The dates given here represent the first deployments. Data temporal coverage will vary by station.
3. Gaps in coverage exist, particularly in Western States.

1413 4. Similar instruments mounted on the NASA Aqua satellite and the European EUMETSAT MetOp series. Taking
1414 into account the similar instrumentation on multiple platforms increases the temporal spatial resolution