# Peer review of "Rain or Snow: Hydrologic Processes, Observations,"

_Hydrology and Earth System Sciences, 2016_

## Referee Comment (RC1) · Anonymous Referee #1 · 7 Oct 2016

This review paper is an essential contribution to the science of understanding precipitation phase and improving our understanding of water resources. I expect this paper will be highly cited and circulated in the community and reviewed by funding program managers and modeling groups interested in improving hydroclimate at various scales (from small-scale hydrologic models to global climate models). This paper summarizes a number of key points and areas for improvements for understanding basic physics and improving our modeling of the hydroclimate. Moreover, the information is well written, presented clearly and concisely, and of high grammatical quality. I only have a few minor suggestions and typo changes to improve the manuscript.

Specific comments:

1) Line 21: Change "The review" to "This review" or "Our review". The previous sen-

tence structure made it unclear which review is being referred to and required the reader to go back to the previous sentence wondering what review is being mentioned.

2) Line 184: either here or elsewhere, it should be mentioned that it is important to validate these microphysics (or other properties if you move this to the discussion) over various land surfaces / types. A microphysics scheme that performs well in Iowa (flat prairie) may not perform well over Idaho (complete mountain terrain) or the Oregon Cascades (coastal warm snow).

3) Line 248: The "(" should be moved to before 1967 based on how the reference is integrated into the sentence

4) Line 303: need parenthesis instead of brackets

5) Lines 433, 601: a space is needed between references

6) Line 583: what is meant by "performing the best"? The best precipitation over mountains? Lowest errors in climatology? Lowest errors in variability? Please clarify.

7) Line 641: "too" not "to"

8) Line 783 and Figure 3: The authors should consider adding an accompanying western U.S. climatology map of humidity to show it has significant spatial variability (implied by the statement here and similar ones elsewhere, but not presently shown).

9) Conclusion/Discussion: I would like to see a paragraph added here or in the previous section (5.6) discussing the implications of this review / points raised for findings from climate change studies focused on snowfall. For example, there are several done at the global scale / continental scale (O'Gorman 2014; Cayan and Pierce. 2013; Kapnick and Delworth 2012). These studies present large-scale changes in snowfall mainly due to temperature (all use temperature-based metrics for phase partitioning), but based on this review, miss the non-temperature induced sensitivity of phase type, likely with nonlinear consequences. Should the changes found in these studies be expected as the temperature signal at some point overwhelms all other signals? Or might the

differences due to climate change be non-linear in all cases? A nice final point of this manuscript would place this study within the framework of these larger scale studies / findings as it is implied that reviewing and exploring phase type will have consequences for understanding future water availability and change.

10) Figure 1: The arrows and curly bracket should be changed to be a different color (not grey) to provide contrast. Perhaps red or blue? They presently do not stand out easily / show the movement of information as presently shown. A more contrasting color choice will make this figure easier to read and understand.

References

Pierce, D.W. and Cayan, D.R., 2013. The uneven response of different snow measures to human-induced climate warming. Journal of Climate, 26(12), pp.4148-4167.

Kapnick, S.B. and Delworth, T.L., 2013. Controls of global snow under a changed climate. Journal of Climate, 26(15), pp.5537-5562.

O'Gorman, P.A., 2014. Contrasting responses of mean and extreme snowfall to climate change. Nature, 512(7515), pp.416-418.

---

## Referee Comment (RC2) · Anonymous Referee #2 · 17 Oct 2016

**1 General comments**

**1.1 Summary of goals, approaches and conclusions**

Harpold et al. wrote a review manuscript on precipitation phase methods (PPM) to show the diversity of available approaches and the advances needed to improve predictions in complex terrain. They reviewed (1) the processes determining the precipitation phase (PP), (2) observation possibilities of the PP and (3) available PPMs. Finally, they formulated research gaps and recommendations on how to improve PPMs. For the second item they concluded that for complex terrain robust observation networks are missing and that for remotely sensed indirect observations of the PP field validations are needed. For different available PPMs they found in their review that accuracy generally increased when humidity is included and when the model is applied on

smaller time steps. They formulated also recommendations for future research based on found research gaps.

**1.2 Conceptual overview**

Up to now most hydrological models are applying primitive PPMs despite the knowledge of atmospheric processes influencing the PP. Thus, this review paper is interesting since it shows the hydrological community the need of future research on this topic. A special emphasis is on the potential on remotely sensed observations. However, especially for the remote sensing part I would like to see a clearer synopsis on how useful these products are for observing the PP in general, additionally to the large amount details listed for each sub method. I would also like to see thoughts and ideas on how the authors would approach their formulated research gaps conceptually, for example on how to include atmospheric models in PPMs. In a review paper I would like to see that the authors are able to create something new with the gathered information. Both mentioned points may be a chance on how to include this aspect. I also suggest that the authors include more details in the manuscript referring to their emphasis on complex terrain in the abstract. There are some formality issues, for example when citing literature. Since this is a review paper, I gave special attention to this aspect. In general, I think that the authors can address my comments with minor revisions. Please find in the following sections details to my above mentioned critics.

**2 Specific comments**

**2.1 Synopsis of remotely sensed information**

Section 3.2 and 3.3 are quite long indicating an emphasis on remotely sensed observations. After reading the two sections I feel that a synopsis is missing with general information about the applicability of those observations for PPM, which seems a bit lost in the detailed description in these long sections. I would suggest a summarizing paragraph, or an overview table with the following items, for example: description, coverage, availability, resolution, validated, references. The remotely sensed observations
do also hardly appear in section 5 (Research Gaps), while the need to validate these products, was mentioned in the abstract. This synopsis can also be placed in the very short Conclusion section, in which the remotely sensed observations are also only very briefly mentioned (line 800).

2.2 Incorporation of atmospheric information

The authors describe well in section 4.2 the problematic scale issue between kilometer-scaled atmospheric models and processes influencing PP which act on a finer resolution. They emphasize that "...grid cells are averages requiring hydrological modellers to consider effects of elevation, aspect, etc. in resolving precipitation phase fractions for finer-scaled models." (l588ff). I think this is a very relevant topic and I would like to see this topic further discussed in the research gap section, maybe even with some conceptual ideas and/or reference to existing work, or – if not existent – references to similar work done by the downscaling community to represent unresolved variability on the sub-grid scale.

The authors also promote in section 5.5 (Develop spatially resolved products) the benefit of gridded products. Since these products probably suffer the same scale problems as mentioned in l588 for atmospheric models, the authors may discuss this aspect of including sub-grid variability here as well.

2.3 Specific conclusions for complex terrain

The authors mention in the abstract that the manuscript "...conveys the advancements needed to improve predictions in complex terrain..." (l22f) and that in complex terrain robust observation networks are missing (l26f). I cannot find many details in the manuscript which allow formulating such a focus on complex terrain in the abstract. I suggest adding a paragraph in the research gap section summarizing specific issues in complex terrain.

2.4 Formality issues

I would in general like to see page numbers to relevant sections when citing a book (or similar). One prominent example is the book authored by the U.S. Army Corps of Engineers, which regularly is available as a non-searchable pdf document or as a hardcopy. It contains various topics relevant to snow hydrology. To find the cited paragraph without mentioning page numbers is nearly impossible. I think this example shows that the standard of including page numbers when citing books and similar long references should be used. Similarly, the authors have not included access dates for all cited URL (e.g. line 200, line 1077 and others). Some cited references appear different than others (sometimes white spaces between ";" sometimes italic "et al.", sometimes with square brackets). More importantly, there are a few citations which do not appear in the reference list. These points are mentioned in my section "Comments line by line" below.

2.5 Motivate Figures in the text

Figure 1 and Figure 4 are hardly described in the text, although containing important information. I would suggest that the authors link their text closer to those Figures, especially to Figure 1 which shows the consequences of wrong PP in a hydrological model.

2.6 Explain abbreviations and lines in Figure 2

It is not clear to me what the blue dotted line is (probably the mixing ratio). I would also suggest to add the used abbreviations for H, LE, f(sat), r etc in the caption. The arrow after H or LE should probably indicate that the energy of the hydrometeor is increasing because of a sensible heat transfer? Please clarify these uncertainties.

3 Comments line-by-line

Line 33ff: This sentence is the same as the previous.

Line 200/208: Please use access dates with URLs. I suggest putting the links in the reference list.

Line 231: Lejeune not in reference list.

Line 265. Not clear which the comparison study is.

Line 354. The cited study is called Arkin and Ardanuy (1998).

Line 411 and elsewhere: Kulie and Bennartz (2003) not in reference list

Line 539: Froidurot wrongly spelled.

Line 945: no page numbers

Line 973: Krug (1995) and Bergström (1995) refer to the same document

Line 978: Missing page numbers

Line 1037/1040: Please use McCabe and Wollock (1999a) and (1999b)

Line 1213: two times YE et al. (2013) in reference list

Line 1178: delete "publication info" and add page numbers

Table 1: McCabe and Wollock (2009) not in reference list.

---

## Author Comment (AC1) · 10 Nov 2016

We appreciate the reviewer's positive overall comments on the manuscript. We make detailed responses below below and have made the editorial changes wherever possible. We also agree with the more major comments regarding validation of the microphysics schemes and the potential influence of temperature-only PPM in large-scale forecasting of phase under changing climate.

Response to specific comments:

1) Line 21: Change "The review" to "This review" or "Our review". The previous sentence structure made it unclear which review is being referred to and required the reader to go back to the previous sentence wondering what review is being mentioned.

[Figure]

This was changed. We also made similar changes on line 114 and 795.

2) Line 184: either here or elsewhere, it should be mentioned that it is important to validate these microphysics (or other properties if you move this to the discussion) over various land surfaces / types. A microphysics scheme that performs well in Iowa (flat prairie) may not perform well over Idaho (complete mountain terrain) or the Oregon Cascades (coastal warm snow).

This was expanded on line 183 to read "The rain-snow line predicted by atmospheric models is very sensitive to these microphysics (Minder, 2010) and validating these microphysics across locations with complex physiography is challenging."

We agree that a discussion of verifying the microphysics schemes, in particular for the complex terrain that is the focus of the paper, would strengthen the paper. We have a added several sentences beginning on line 602-609: "These schemes vary greatly in their accuracy with "mixed phase" schemes generally having the best verifying simulations of precipitation in complex terrain where much of the water is supercooled (Lin, 2007; Reisner et al., 1998; Thompson et al., 2004; Thompson et al., 2008; Morrison et al., 2005; Zängl, 2007; Kaplan et al., 2012). Comprehensive validation of the microphysics schemes over different land surfaces types with a focus on different snowfall patterns (e.g. warm maritime, flat prairie, etc.) is lacking. In particular, in transition zones between mountains and plains or along coast lines the complexity of the microphysics becomes even more extreme as differing air mass characteristics become juxtaposed.

3) Line 248: The "(" should be moved to before 1967 based on how the reference is integrated into the sentence

This was corrected.

4) Line 303: need parenthesis instead of brackets

This was corrected.

5) Lines 433, 601: a space is needed between ween references

This was corrected.

6) Line 583: what is meant by "performing the best"? The best precipitation over mountains? Lowest errors in climatology? Lowest errors in variability? Please clarify. This was clarified on line 616: "These schemes vary greatly in their accuracy with "mixed phase" schemes generally having the best verifying simulations of precipitation in complex terrain where much of the water is supercooled (Lin, 2007; Reisner et al., 1998; Thompson et al., 2004; Thompson et al., 2008; Morrison et al., 2005; Zängl, 2007; Kaplan et al., 2012)."

7) Line 641: "too" not "to"

This was corrected.

8) Line 783 and Figure 3: The authors should consider adding an accompanying western U.S. climatology map of humidity to show it has significant spatial variability (implied by the statement here and similar ones elsewhere, but not presently shown).

This is a good suggestion. We will add a map as a second panel to Figure 3 using the University of Idaho Gridded Meteorological Datasets, which is essentially NLDAS-2 data downscaled to 4 km.

9) Conclusion/Discussion: I would like to see a paragraph added here or in the previous section (5.6) discussing the implications of this review / points raised for findings from climate change studies focused on snowfall. For example, there are several done at the global scale / continental scale (O'Gorman 2014; Cayan and Pierce. 2013; Kapnick and Delworth 2012). These studies present large-scale changes in snowfall mainly due to temperature (all use temperature-based metrics for phase partitioning), but based on this review, miss the non-temperature induced sensitivity of phase type, likely with nonlinear consequences. Should the changes found in these studies be expected as the temperature signal at some point overwhelms all other signals? Or might the

differences due to climate change be non-linear in all cases? A nice final point of this manuscript would place this study within the framework of these larger scale studies / findings as it is implied that reviewing and exploring phase type will have consequences for understanding future water availability and change.

This is an excellent point that was hinted at in the discussion but not fully addressed. We expanded section 5.6 on line 861 to read: "Because broad-scale techniques applied to assess changes in precipitation phase and snowfall have relied on temperature, both regionally (Klos et al., 2014; Pierce and Cayan, 2013; Knowles et al., 2006) and globally (Kapnick and Delworth, 2013; O'Gorman, 2014), they have not fully considered the potential non-linearaties created by the absence of wet bulb depressions and humidity in assessment of sensitivity to changes in phase."

10) Figure 1: The arrows and curly bracket should be changed to be a different color (not grey) to provide contrast. Perhaps red or blue? They presently do not stand out easily / show the movement of information as presently shown. A more contrasting color choice will make this figure easier to read and understand.

We agree and have made these changes to a new figure that is attached.

[Figure]

**over predict RAIN proportion:**

intercep-tion

surface runoff

snow storage

infiltration

[Figure]

**over predict SNOW proportion:**

intercep-tion

surf. runoff

infil-tration

snow storage

water more in

---

## Author Comment (AC2) · 10 Nov 2016

We appreciate the reviewer's constructive and specific comments on the manuscript. We have addressed all the minor editorial comments and responded to the more detailed comments in the text below. We agree that the sign of a good review paper is creating something new from the gathered information, which was the objective of Section 5. We will bolster that effort by following the reviewer's recommendation about more details on the incorporation of atmospheric models into PPM and better explaining the importance/role of complex terrain.

2 Specific comments

2.1 Synopsis of remotely sensed information Section 3.2 and 3.3 are quite long indicating an emphasis on remotely sensed observations. After reading the two sections I

feel that a synopsis is missing with general information about the applicability of those observations for PPM, which seems a bit lost in the detailed description in these long sections. I would suggest a summarizing paragraph, or an overview table with the following items, for example: description, coverage, availability, resolution, validated, references. The remotely sensed observations do also hardly appear in section 5 (Research Gaps), while the need to validate these products, was mentioned in the abstract. This synopsis can also be placed in the very short Conclusion section, in which the remotely sensed observations are also only very briefly mentioned (line 800).

We agree that a reader could get lost in the details of this section and not see the bigger picture. To improve this section, we have added both a brief overview at the beginning of section 3.2 and 3.3, as well as a table that more succinctly summarizes the technologies. The research gaps section did discuss remote sensing in section 5.2 and 5.5. Section 5.6 had the following sentence added on line 582: "Recent remote sensing platforms, such as GPM, may offer an additional tool to assess regional variability, however, the current GPM precipitation phase product relies on wet bulb temperatures based on model output and not microwave-based observations (Huffman et al., 2015)."

We also acknowledged that not only the PPM algorithms need improvement but also observations from remote sensing in the conclusions. The first paragraph of section 3.2 now reads "Ground-based remote sensing observations have been available for several decades to detect precipitation phase using bright band heights. Until recently, most ground-based radar stations were operated as conventional Doppler systems that transmit and receive radio waves with single horizontal polarization. Developments in dual polarization ground radar such as those that function as part of the U.S. National Weather Service NEXRAD network, have resulted in systems that transmit radio signals with both horizontal and vertical polarizations. In general, ground-based remote sensing observation, either single or dual-pol, remain underutilized for detecting precipitation phase and are challenging to apply in complex terrain (Table 2)."

The first paragraph is Section 3.3 now reads "Spaceborne remote sensing observations typically use passive or active microwave sensors to determine precipitation phase (Table 2). While many of the previous passive microwave systems were challenged by coarse resolutions and difficulties retrieving snowfall over snow-covered areas. More recent active microwave systems have advantage for detecting phase in terms of accuracy and spatial resolution, but remain largely unverified. Table 2 provides and overview of these space-based remote sensing technologies that are described in more detail below."

The table has information on single polarized and dual-polarized ground radar, and spaceborne passive and active microwave sensors. The information in the table will include description, spatial resolution, temporal resolution, phase validation, and relevant references.

2.2 Incorporation of atmospheric information The authors describe well in section 4.2 the problematic scale issue between kilometer- scaled atmospheric models and processes influencing PP which act on a finer resolution. They emphasize that ". . .grid cells are averages requiring hydrological modellers to consider effects of elevation, aspect, etc. in resolving precipitation phase fractions for finer-scaled models." (l588ff). I think this is a very relevant topic and I would like to see this topic further discussed in the research gap section, maybe even with some conceptual ideas and/or reference to existing work, or – if not existent – references to similar work done by the downscaling community to represent unresolved variability on the sub-grid scale.

We agree that model scale is an important effect to consider and have added text to section 5.2 staring on line 750: "Historically, meteorological models have not been run at spatial scales capable of resolving convection (e.g. <2 km), which can exacerbate error in precipitation phase in complex terrain with a moisture neutral atmosphere. Coarse meteorological models also struggle to produce pockets of frozen precipitation from advection of moisture plumes between mountain ranges and cold air wedged between barriers. However, reduced computational restrictions on running these models at finer spatial-scales and over large geographic extents (Rasmussen et al., 2012) are

enabling further investigations into precipitation phase change under historical and future climate scenarios. The suggests that finer dynamical downscaling is necessary to resolve precipitation phase is consistent with similar work attempting to resolve winter precipitation amount in complex terrain (Gutmann et al., 2014). A potentially impactful area of research is to integrate this information into novel approaches to improve precipitation phase prediction skill."

The authors also promote in section 5.5 (Develop spatially resolved products) the benefit of gridded products. Since these products probably suffer the same scale problems as mentioned in l588 for atmospheric models, the authors may discuss this aspect of including sub-grid variability here as well.

We agree and add this sentence in section 5.5 beginning on line 847: "Accurate gridded phase products rely on the ability to represent the physics of water vapor and energy flows in complex terrain (e.g. Holden et al., 2010) where statistical downscaling methods are typically insufficient (Gutmann et al., 2012)."

2.3 Specific conclusions for complex terrain The authors mention in the abstract that the manuscript ". . .conveys the advancements needed to improve predictions in complex terrain..." (l22f) and that in complex terrain robust observation networks are missing (l26f). I cannot find many details in the manuscript which allow formulating such a focus on complex terrain in the abstract. I suggest adding a paragraph in the research gap section summarizing specific issues in complex terrain.

The reviewer makes an important point that we address in numerous places within the manuscript. On line 188: "The rain-snow line predicted by atmospheric models is very sensitive to these microphysics (Minder, 2010) and validating these microphysics across locations with complex physiography is challenging." Line 203: "Few research stations, however, have this benefit, particularly in many remote regions and in complex terrain." On line 244: "However, if these observation systems were sufficiently simple they may have the potential to be applied operationally across larger meteorological monitoring networks encompassing complex terrain where snow comprises a large component of annual precipitation (Rajagopal and Harpold, 2016)." On line 298: "In general, ground-based remote sensing observation, either single or dual-pol, remain underutilized for detecting precipitation phase and are challenging to apply in complex terrain (Table 2)." On line 616: "These schemes vary greatly in their accuracy with "mixed phase" schemes generally having the best verifying simulations of precipitation in complex terrain where much of the water is supercooled (Lin, 2007; Reisner et al., 1998; Thompson et al., 2004; Thompson et al., 2008; Morrison et al., 2005; Zängl, 2007; Kaplan et al., 2012). Comprehensive validation of the microphysics schemes over different land surfaces types with a focus on different snowfall patterns (e.g. warm maritime, flat prairie, etc.) is lacking. In particular, in transition zones between mountains and plains or along coast lines the complexity of the microphysics becomes even more extreme as differing air mass characteristics become juxtaposed."

We add a new paragraph in at the beginning of the research gap section: "Similar intensive field campaigns are needed in complex terrain that is frequently characterized by highly dynamic and spatially variable hydrometeorological conditions. Such campaigns are expensive to conduct, but can be implemented as part of operational nowcasting to develop rich data resources to advance scientific understanding as was very effectively done during the Vancouver Olympic Games in 2010 (Isaac et al., 2014; Joe et al., 2014). The research community should capitalize on similar opportunities and expand environmental monitoring networks to simultaneously advance both atmospheric and hydrological understanding, especially in complex terrain spanning the rain-snow transition zone." We also add this sentence to section 5.1: "In complex terrain, air temperature can also vary dramatically at relatively small scales from ridgetops to valley bottoms due to cold air drainage (Whiteman et al., 1999) and hence can introduce errors into inferential techniques such as these." Multiple sentences are added to section 5.2: "Historically, meteorological models have not been run at spatial scales capable of resolving convection (e.g. <2 km), which can exacerbate error in precipitation phase in complex terrain with a moist neutral atmosphere. Coarse meteorological models also

struggle to produce pockets of frozen precipitation from advection of moisture plumes between mountain ranges and cold air wedged between barriers. However, reduced computational restrictions on running these models at finer spatial-scales and over large geographic extents (Rasmussen et al., 2012) are enabling further investigations into precipitation phase change under historical and future climate scenarios. This suggests that finer dynamical downscaling is necessary to resolve precipitation phase which is consistent with similar work attempting to resolve winter precipitation amount in complex terrain (Gutmann et al., 2014)." And an additional sentence in section 5.5: "Accurate gridded phase products rely on the ability to represent the physics of water vapor and energy flows in complex terrain (e.g. Holden et al., 2010) where statistical downscaling methods are typically insufficient (Gutmann et al., 2012)."

2.4 Formality issues I would in general like to see page numbers to relevant sections when citing a book (or similar). One prominent example is the book authored by the U.S. Army Corps of Engineers, which regularly is available as a non-searchable pdf document or as a hardcopy. It contains various topics relevant to snow hydrology. To find the cited paragraph without mentioning page numbers is nearly impossible. I think this example shows that the standard of including page numbers when citing books and similar long references should be used. Similarly, the authors have not included access dates for all cited URL (e.g. line 200, line 1077 and others). Some cited references appear different than others (sometimes white spaces between ";" sometimes italic "et al.", sometimes with square brackets). More importantly, there are a few citations which do not appear in the reference list. These points are mentioned in my section "Comments line by line" below.

We appreciate the reviewer's attention to detail and have corrected these in the text and references.

2.5 Motivate Figures in the text Figure 1 and Figure 4 are hardly described in the text, although containing important information. I would suggest that the authors link their text closer to those Figures, especially to Figure 1 which shows the consequences of

wrong PP in a hydrological model.

This is a good point by the reviewer. We add additional references to Figure 1 in the introduction. We also add this sentence to the beginning of section 5: "The cascading effects of incorrectly predicting precipitation phase lead to cascading effects on hydrological modeling (Figure 1)." We also better reference Figure 4 at the beginning of section 5 and within section 5.4.

2.6 Explain abbreviations and lines in Figure 2

It is not clear to me what the blue dotted line is (probably the mixing ratio). I would also suggest to add the used abbreviations for H, LE, f(sat), r etc in the caption. The arrow after H or LE should probably indicate that the energy of the hydrometeor is increasing because of a sensible heat transfer? Please clarify these uncertainties.

The following lines have been added to the caption for figure 2: "The blue dotted line represents the mixing ratio. H, LE, f(sat), and r are abbreviations for sensible heat, latent heat of evaporation, function of saturation and mixing ratio respectively. The arrow after H or LE indicate the energy of the hydrometeor either increasing (up) or decreasing (down) which is controlled by other atmospheric conditions."

3 Comments line-by-line

Line 33ff: This sentence is the same as the previous.

This was deleted.

Line 200/208: Please use access dates with URLs. I suggest putting the links in the reference list.

This was corrected throughout the document.

Line 231: Lejeune not in reference list. This was incorrect and changed to L'hôte et al., 2005.

Line 265. Not clear which the comparison study is.

This was corrected to read: In a comparison study by Caraccioloa et al., (2006), the PARSIVEL optical disdrometer, originally described by Loffler-Mang et al. (1999) did not perform well against a 2DVD because of problems related to the detection of slow fall velocities for snow.

Line 354. The cited study is called Arkin and Ardanuy (1998).

This was corrected.

Line 411 and elsewhere: Kulie and Bennartz (2003) not in reference list

This was corrected.

Line 539: Froidurot wrongly spelled.

This was corrected

Line 945: no page numbers

This was corrected

Line 973: Krug (1995) and Bergström (1995) refer to the same document .

This was corrected

Line 978: Missing page numbers

This was corrected

Line 1037/1040: Please use McCabe and Wollock (1999a) and (1999b)

This was corrected

Line 1213: two times YE et al. (2013) in reference list

This was corrected

Line 1178: delete "publication info" and add page numbers

This was corrected

Table 1: McCabe and Wollock (2009) not in reference list.

This was corrected

---

## Author Response (AR1)

[revised manuscript text omitted]

Adrian Harpold 11/20/16 2:22 PM Deleted: as snow or rain controls numerous Adrian Harpold 11/20/16 2:22 PM Deleted: are fundamental to effective Adrian Harpold 11/20/16 2:22 PM Deleted: ing Adrian Harpold 11/20/16 2:22 PM Deleted: its Adrian Harpold 11/20/16 2:22 PM Deleted: use overly simplistic estimates Adrian Harpold 11/20/16 2:22 PM Deleted: The Adrian Harpold 11/20/16 2:22 PM Deleted: characterized by Adrian Harpold 11/20/16 2:22 PM Deleted: the Adrian Harpold 11/20/16 2:22 PM Deleted: underlying Adrian Harpold 11/20/16 2:22 PM **Deleted:** the types and accuracy of Adrian Harpold 11/20/16 2:22 PM Deleted: to show Adrian Harpold 11/20/16 2:22 PM Deleted: time steps Adrian Harpold 11/20/16 2:22 PM Deleted: Adrian Harpold 11/20/16 2:22 PM Deleted: offers numerous models and Adrian Harpold 11/20/16 2:22 PM Deleted: 
[revised manuscript text omitted]

Adrian Harpold 11/20/16 2:22 PM Deleted: point Adrian Harpold 11/20/16 2:22 PM Deleted: knowledge

Adrian Harpold 11/20/16 2:22 PM Deleted: This

Adrian Harpold 11/20/16 2:22 PM Deleted: is typically Adrian Harpold 11/20/16 2:22 PM Deleted: as

Adrian Harpold 11/20/16 2:22 PM Deleted: < Adrian Harpold 11/20/16 2:22 PM Deleted: ( Adrian Harpold 11/20/16 2:22 PM Deleted: ), Adrian Harpold 11/20/16 2:22 PM Deleted: (T >= Adrian Harpold 11/20/16 2:22 PM Deleted: )

| 192 | critical height. | The temperature | profile and de | pth of the surface la | iver controls the | precipitation |
|-----|------------------|-----------------|----------------|-----------------------|-------------------|---------------|
|     |                  |                 |                |                       | 1                 |               |

- 193 phase reaching the ground surface. For example, in Figure 2a, if rain reaches the critical height, it
- 194 may reach the surface as rain or ice pellets depending on small differences in temperature in the
- surface layer (Theriault and Stewart, 2010). Similarly, in Figure 2b, if snow reaches the critical
- 196 | height, it may reach the surface as snow if the temperature in the surface layer is below freezing.
- 197 However, in Figure 2c, when the surface layer temperatures are close to freezing and the mixing
- ratios are neither close to saturation or very dry the phase at the surface is not easily determined

199 by the surface conditions alone.

200

In addition to strong dependence on the vertical temperature and humidity profiles, precipitation 201 phase is also a function of fall rate and hydrometeor size because they affect energy exchange 202 203 with the atmosphere (Theriault et al., 2010). Precipitation rate influences the precipitation phase; for example, a precipitation rate of 10 mm h-1 reduces the amount of freezing rain by a factor of 204 three over a precipitation rate of 1 mm h-1 (Theriault and Stewart, 2010) because there is less 205 time for exchange of turbulent heat with the hydrometeor. A solid hydrometeor that originates in 206 the top layer and falls through the mixed layer can reach the surface layer as wet snow, sleet, or 207 208 rain. This phase transition in the mixed layer is primarily a function of latent heat exchange driven by vapor pressure gradients and sensible heat exchange driven by temperature gradients. 209 210 Temperature generally increases from the mixed layer to the surface layer causing sensible heat 211 inputs to the hydrometeor. If these gains in sensible heat are combined with minimal latent heat 212 losses resulting from low vapor pressure deficits, it is likely the hydrometeor will reach the 213 surface layer as rain (Figure 2). However, vapor pressure in the mixed layer is often below saturation leading to latent energy losses and cooling of the hydrometeor coupled with diabatic 214 cooling of the local atmosphere, which can produce snow or other forms of frozen precipitation 215 at the surface even when temperatures are above  $0_{\bullet}^{\circ}$  C. Likewise, surface energetics affect local 216 atmospheric conditions and dynamics, especially in complex terrain. For example, melting of the 217 snowpack can cause diabatic cooling of the local atmosphere and affect the phase of 218 precipitation, especially when air temperatures are very close to 0 °C (Theriault et al., 2012). 219 220 Many conditions lead to a combination of latent heat losses and sensible heat gains by 221 hydrometeors (Figure 2). Under these conditions it can be difficult to predict the phase of

Adrian Harpold 11/20/16 2:22 PM Deleted: since

Adrian Harpold 11/20/16 2:22 PM Deleted: °

Adrian Harpold 11/20/16 2:22 PM Deleted: of

| 225 | precipitation | without suf | ficient infor | mation about | humidity an | id tempera | ture profiles, | turbulence, |
|-----|---------------|-------------|---------------|--------------|-------------|------------|----------------|-------------|
|-----|---------------|-------------|---------------|--------------|-------------|------------|----------------|-------------|

- 226 hydrometeor size, and precipitation intensity.
- 227

[revised manuscript text omitted]

Adrian Harpold 11/20/16 2:22 PM Deleted: .

Adrian Harpold 11/20/16 2:22 PM Deleted: incorporated Adrian Harpold 11/20/16 2:22 PM Deleted: 2

Adrian Harpold 11/20/16 2:22 PM Deleted: (http://www.nws.noaa.gov/om/coop/ what-is-coop.html

| 262 | network of volunteers recording daily observations of temperature and precipitation, including     |
|-----|----------------------------------------------------------------------------------------------------|
| 263 | phase. The NOAA National Severe Storms Laboratory used citizen scientist observations of rain      |
| 264 | and snow occurrence to evaluate the performance of the Multi-Radar Multi-Sensor (MRMS)             |
| 265 | system in the meteorological Phenomena Identification Near the Ground (mPING) project (Chen        |
| 266 | et al., 2015). The Colorado Climate Center initiated Community Collaborative Rain, Hail and        |
| 267 | Snow Network (CoCoRaHS) supplies volunteers with low cost instrumentation to observe               |
| 268 | precipitation characteristics, including phase, and enables observations to be reported on the     |
| 269 | project website (http://www.cocorahs.org/, accessed 10/12/2016). Although highly valuable,         |
| 270 | some limitations of this system include the imperfect ability of observers to identify mixed phase |
| 271 | events and the temporal extent of storms, as well as the lack of observations in both remote areas |
| 272 | and during low light conditions.                                                                   |
| 273 |                                                                                                    |
| 274 | Coupled observations link synchronous measurements of precipitation with secondary                 |
| 275 | observations to indicate phase. Secondary observations can include photographs of surrounding      |
| 276 | terrain, snow depth measurements, and measurements of ancillary meteorological variables.          |
| 277 | Photographs of vertical scales emplaced in the snow have been used to estimate snow                |
| 278 | accumulation depth, which can then be coupled with precipitation mass to determine density and     |
| 279 | phase (Berris and Harr, 1987; Floyd and Weiler, 2008; Garvelmann et al., 2013; Hedrick and         |
| 280 | Marshall, 2014; Parajka et al., 2012). Mixed phase events, however, are difficult to quantify      |
| 281 | using coupled depth- and photographic-based techniques (Floyd and Weiler, 2008). Acoustic          |
| 282 | distance sensors, which are now commonly used to monitor the accumulation of snow (e.g. Boe,       |
| 283 | 2013), have similar drawbacks in mixed phase events, but have been effectively applied to          |
| 284 | separate snow from rain (Rajagopal and Harpold, 2016). Meteorological information such as          |
| 285 | temperature and relative humidity can be used to compute the phase of precipitation measured by    |
| 286 | bucket-type gauges. Unfortunately, this approach generally requires incorporating assumptions      |
| 287 | about the meteorological conditions that determine phase (see section 4.1). Harder and Pomeroy     |
| 288 | (2013) used a comprehensive approach to determine the phase of precipitation. Every 15 minutes     |
| 289 | during their study period phase was determined by evaluating weighing bucket mass, tipping         |
| 290 | bucket depth, albedo, snow depth, and air temperature. Similarly, Marks et al. (2013) used a       |
| 291 | scheme based on co-located precipitation and snow depth to discriminate phase. A more              |
| 292 | involved expert decision making approach by L'hôte et al. (2005) was based on six recorded  |

Adrian Harpold 11/20/16 2:22 PM Deleted: (http://www.cocorahs.org/).

Adrian Harpold 11/20/16 2:22 PM Deleted: Lejeune Adrian Harpold 11/20/16 2:22 PM Deleted: 3

meteorological parameters: precipitation intensity, albedo of the ground, air temperature, ground 296 surface temperature, reflected long-wave radiation, and soil heat flux. The intent of most of these 297 298 coupled observations was to develop datasets to evaluate PPM algorithms. However, if these 299 observation systems were sufficiently simple they may have the potential to be applied operationally across larger meteorological monitoring networks encompassing complex terrain 300 301 where snow comprises a large component of annual precipitation (Rajagopal and Harpold, 2016). 302 Proxy observations measure geophysical properties of precipitation to infer phase. The hot plate 303 304 precipitation gauge introduced by Rasmussen et al. (2012), for example, uses a heated thin disk to accumulate precipitation and then measures the amount of energy required to melt snow or 305 evaporate liquid water. This technique, however, requires a secondary measurement of air 306 307 temperature to determine if the energy is used to melt snow or only evaporate rain. Disdrometers measure the size and velocity of hydrometeors. Although the most common application of 308 disdrometer data is to determine the drop size distribution (DSD) and other properties of rain, the 309 310 phase of hydrometeors can be inferred by relating velocity and size to density. Some disdrometer technologies, which can be grouped into impact, imaging, and scattering approaches (Loffler-311 312 Mang et al., 1999), are better suited for describing snow than others. Impact disdrometers, first introduced by Joss and Waldvogel (1967), use an electromechanical sensor to convert the 313 314 momentum of a hydrometeor into an electric pulse. The amplitude of the pulse is a function of 315 drop diameter. Impact disdrometers have not been commonly used to measure solid precipitation due to the different functional relationships between drop size and momentum for solid and 316 317 liquid precipitation. Imaging disdrometers use basic photographic principles to acquire images of the distribution of particles (Borrmann and Jaenicke, 1993; Knollenberg, 1970). The 2D Video 318 Disdrometer (2DVD) described by Kruger and Krajewski (2002) records the shadows cast by 319 320 hydrometeors onto photodetectors as they pass through two sheets of light. The shape of the shadows enables computation of particle size, and shadows are tracked through both light sheets 321 to determine velocity. Although initially designed to describe liquid precipitation, recent work 322 has shown that the 2DVD can be used to classify snowfall according to microphysical properties 323 324 of single hydrometeors (Bernauer et al., 2016). The 2DVD has been used to classify known rain 325 or snow events individually, but little work has been performed to distinguish between liquid and solid precipitation. Scattering disdrometers, or optical disdrometers, measure the extinction of 326

Adrian Harpold 11/20/16 2:22 PM Deleted: soil

Adrian Harpold 11/20/16 2:22 PM Deleted: ( Adrian Harpold 11/20/16 2:22 PM Deleted: ,

| 330 | light passing between a source and a sensor (Hauser et al., 1984; Loffler-Mang et al., 1999). Like  |
|-----|-----------------------------------------------------------------------------------------------------|
| 331 | the other types, optical disdrometers were originally designed for rain, but have been periodically |
| 332 | applied to snow (Battaglia et al., 2010; Lempio et al., 2007). In a comparison study, by            |
| 333 | Caracciolo et al. (2006), the PARSIVEL optical disdrometer, originally described by Loffler-        |
| 334 | Mang et al. (1999) did not perform well against a 2DVD because of problems related to the           |
| 335 | detection of slow fall velocities for snow. It may be possible to use optical disdrometers to       |
| 336 | distinguish between rain, sleet, and snow based on the existence of distinct shapes of the size     |
| 337 | spectra for each precipitation type. More research on the relationship between air temperature      |
| 338 | and the size spectra produced by the optical disdrometer is needed (Lempio et al., 2007). In        |
| 339 | summary, disdrometers of various types are valuable tools for describing the properties of rain     |
| 340 | and snow, but require further testing and development to distinguish between rain and snow, as      |
| 341 | well as mixed phase events.                                                                         |
| 342 |                                                                                                     |
| 343 | 3.2 Ground-based remote sensing observations                                                        |
| 344 | Ground-based remote sensing observations have been available for several decades to detect          |
| 345 | precipitation phase using radar. Until recently, most ground-based radar stations were operated     |
| 346 | as conventional Doppler systems that transmit and receive radio waves with single horizontal        |
| 347 | polarization. Developments in dual polarization ground radar such as those that function as part    |
| 348 | of the U.S. National Weather Service NEXRAD network (NOAA, 2016), have resulted in                  |
| 349 | systems that transmit radio signals with both horizontal and vertical polarizations. In general,    |
| 350 | ground-based remote sensing observation, either single or dual-pol, remain underutilized for        |
| 351 | detecting precipitation phase and are challenging to apply in complex terrain (Table 2).            |
| 352 |                                                                                                     |
| 353 | Ground-based remote sensing of precipitation phase using single-polarized radar systems             |
| 354 | depends on detecting the radar bright band. Radio waves transmitted by the radar system, are        |
| 355 | scattered by hydrometeors in the atmosphere, with a certain proportion reflected back towards       |
| 356 | the radar antenna. The magnitude of the measured reflectivity (Z) is related to the size and the    |
| 357 | dielectric constant of falling hydrometeors (White et al., 2002). Ice particles aggregate as they   |
| 358 | descend through the atmosphere and their dielectric constant increases, in turn increasing Z        |
| 359 | measured by the radar, creating the bright band, a layer of enhanced reflectivity just below the    |
| 360 | elevation of the melting level (Lundquist et al., 2008). Therefore, bright band elevation can be    |

Adrian Harpold 11/20/16 2:22 PM Deleted: (Battaglia et al., 2010; Lempio et al., 2007). Adrian Harpold 11/20/16 2:22 PM Deleted: ,

Adrian Harpold 11/20/16 2:22 PM Deleted: , Adrian Harpold 11/20/16 2:22 PM Deleted: This section will review techniques Adrian Harpold 11/20/16 2:22 PM Deleted: rm Adrian Harpold 11/20/16 2:22 PM Deleted: in

Adrian Harpold 11/20/16 2:22 PM Deleted: used with data from single-Adrian Harpold 11/20/16 2:22 PM Deleted: dual-pol systems.

370 used as a proxy for the "snow level", the bottom of the melting layer where falling snow

transforms to rain (White et al., 2010; White et al., 2002).

372

| 373 |   | Doppler vertical velocity (DVV) is another variable that can be estimated from single-polarized      |  |
|-----|---|------------------------------------------------------------------------------------------------------|--|
| 374 | I | vertically profiling radar, DVV gives an estimate of the velocity of falling particles; as           |  |
| 375 | I | snowflakes melt and become liquid raindrops, the fall velocity of the altered hydrometeors           |  |
| 376 |   | increases. When combined with reflectivity profiles, DVV helps reduce false positive detection       |  |
| 377 |   | of the bright band, which may be caused by phenomena other than snow melting to rain (White          |  |
| 378 |   | et al., 2002). First, DVV and Z are combined to detect the elevation of the bottom of the bright     |  |
| 379 |   | band. Then the algorithm searches for maximum Z above the bottom of the bright band and              |  |
| 380 |   | determines that to be the bright band elevation (White et al., 2002). However, a test of this        |  |
| 381 |   | algorithm on data from a winter storm over the Sierra Nevada found root mean square errors of        |  |
| 382 |   | 326 to 457 m compared to ground observations when bright band elevation was assumed to               |  |
| 383 | l | represent the surface transition from snow to rain (Lundquist et al., 2008). Snow levels in          |  |
| 384 | 1 | mountainous areas, however, may also be overestimated by radar profiler estimates if they are        |  |
| 385 |   | unable to resolve spatial variations close to mountain fronts, since snow levels have been noted     |  |
| 386 |   | to persistently drop on windward slopes (Minder and Kingsmill, 2013). Despite the potential          |  |
| 387 |   | errors, the elevation of maximum Z may be a useful proxy variable for snow level in                  |  |
| 388 |   | hydrometeorological applications in mountainous watersheds because maximum Z will always             |  |
| 389 | l | occur below the freezing level (Lundquist et al., 2008; White et al., 2010)                          |  |
| 390 | 1 |                                                                                                      |  |
| 391 |   | Few published studies have explored the value of bright band-derived phase data for hydrologic       |  |
| 392 |   | modeling. Maurer and Mass (2006) compared the melting level from vertically pointing radar           |  |
| 393 |   | reflectivity against temperature-based methods to assess whether the radar approach could            |  |
| 394 |   | improve determination of precipitation phase at the ground level. In that study, the altitude of the |  |
| 395 |   | top of the bright band was detected and applied across the study basin. Frozen precipitation was     |  |
| 396 |   | assumed to be falling in model pixels above the altitude of the melting level and liquid             |  |
| 397 |   | precipitation was assumed to be falling in pixels below the altitude of the melting layer Maurer     |  |
| 398 |   | and Mass, 2006). Maurer and Mass (2006) found that incorporating radar-detected melting layer        |  |
| 399 | • | altitude improved streamflow simulation results. A similar study that used bright band altitude to   |  |
| 400 |   | classify pixels according to surface precipitation type was not as conclusive; bright band altitude  |  |
|     |   |                                                                                                      |  |

Adrian Harpold 11/20/16 2:22 PM Deleted: radar. It is derived from Adrian Harpold 11/20/16 2:22 PM Deleted: s

Adrian Harpold 11/20/16 2:22 PM Deleted: [ Adrian Harpold 11/20/16 2:22 PM Deleted: ].

Adrian Harpold 11/20/16 2:22 PM Deleted: White et al., 2010;

Adrian Harpold 11/20/16 2:22 PM **Deleted:** (Maurer and Mass, 2006).

[revised manuscript text omitted]

Adrian Harpold 11/20/16 2:22 PM **Deleted:** (Liu 2008; Kulie and Bennartz, 2009)

**drian Harpold 11/20/16 2:22 PM**

Adrian Harpold 11/20/16 2:22 PM Deleted: (Wood et al., 2013;Kulie et al., 2016).

551 The launch of the Global Precipitation Mission (GPM) core observatory in February 2014 holds 552 553 promise for the future deployment of operational snow detection products. Building on the 554 success of the Tropical Rainfall Monitoring Mission (TRMM), the GPM core observatory sensors include precipitation radar (DPR) and microwave imager (GMI). The GMI has two 555 millimeter wave channels (166 and 183 GHz) that are specifically designed to detect and retrieve 556 light rain and snow precipitation. These are more advanced than the sensors onboard the TRMM 557 spacecraft and permit better quantification of the physical properties of precipitating particles, 558 particularly over land at middle to high latitudes (Hou et al., 2014). Algorithms for the GPM 559 mission are still under development, and is partly being driven by data collected during the GPM 560 Cold Season Experiment (GCPEx) (Skofronick-Jackson et al., 2015). Using airborne sensors to 561 562 simulate GPM and DPR measurements, one of the questions that the GCPEx hoped to address concerned the potential capability of data from the DPR and GMI to discriminate falling snow 563 from rain or clear air (Skofronick-Jackson et al., 2015). The initial results reported by the GCPEx 564 study echo some of the challenges recognized for ground-based single polarized radar detection 565 of snowfall. The relationship between radar reflectivity and snowfall is not unique. For the GPM 566 mission, it will be necessary to include more variables from dual frequency radar measurements, 567 multiple frequency passive microwave measurements, or a combination of radar and passive 568 569 microwave measurements (Skofronick-Jackson et al., 2015). 570 4. Current Tools for Predicting Precipitation Phase 571 572 4.1 Prediction Techniques from Ground-Based Observations 573 Discriminating between solid and liquid precipitation is often based on a near-surface air temperature threshold (Martinec and Rango, 1986;U.S. Army Corps of Engineers, 1956;L'hôte et 574 575 al., 2005). Four prediction methods have been developed that use near-surface air temperature for discriminating precipitation phase: 1) static threshold, 2) linear transition, 3) minimum and 576 maximum temperature, and 4) sigmoidal curve (Table 1). A static temperature threshold applies 577 a single temperature value, such as mean daily temperature, where all of the precipitation above 578 579 the threshold is rain, and all below that threshold is snow. Typically this threshold temperature is near 0 °C (Lynch-Stieglitz, 1994; Motoyama, 1990), but was shown to be highly variable across 580 both space and time (Kienzle, 2008; Motoyama, 1990; Braun, 1984; Ye et al., 2013). For 581

Adrian Harpold 11/20/16 2:22 PM Deleted: (Motoyama, 1990; Lynch-Stieglitz, 1994) Adrian Harpold 11/20/16 2:22 PM Deleted: a

example, Rajagopal and Harpold (2016) optimized a single temperature threshold at Snow 585 Telemetry (SNOTEL) sites across the western U.S. to show regional variability from -4 to 3 °C 586 587 (Figure 3). A second discrimination technique is to linearly scale the proportion of snow and rain 588 between a temperature for all rain ( $T_{rain}$ ) and a temperature for all snow ( $T_{snow}$ ) (Pipes and Quick, 1977; McCabe and Wolock, 2010; Tarboton et al., 1995). Linear threshold models have been 589 parameterized slightly differently across studies, e.g.: Tsnow =-1.0 °C, Train = 3.0 °C (McCabe and 590 Wolock, 2010), Tsnow =-1.1 °C and Train =3.3 °C (Tarboton et al., 1995), and Tsnow =0 °C and Train 591 =5 °
[revised manuscript text omitted]

**Adrian Harpold 11/20/16 2:22 PM Deleted: (Yamazaki, 2001).**

Adrian Harpold 11/20/16 2:22 PM Deleted: Ye et al. (2013b) Adrian Harpold 11/20/16 2:22 PM Deleted: Frudoiret et al. (2014)

- 680
- 4.2 Prediction Techniques Incorporating Atmospheric Information 681 682 While many hydrologic models have their own formulations for determining precipitation phase 683 at the ground, it is also possible to initialize hydrologic models with precipitation phase fraction, intensity, and volume from numerical weather simulation model output. Here we discuss the 684 limitations of precipitation phase simulation inherent to WRF (Kaplan et al., 2012; Skamarock et 685 al., 2008) and other atmospheric simulation models. The finest scale spatial resolution employed 686 in atmospheric simulation models is  $\sim 1$  km and these models generate data at hourly or finer 687 688 temporal resolutions. Regional climate models (RCM) and global climate models (GCM) are typically coarser than local mesoscale models. The physical processes driving both the removal 689 of moisture from the air and the precipitation phase (Section 2) occur at much finer spatial and 690 691 temporal resolutions in the real atmosphere than models typically resolve, i.e. <1 km. As with all numerical models, the representation of sub-grid scale processes requires parameterization. At 692 typical scales considered, characterization of mixed phase processes within a condensing cloud 693 depends on both cloud microphysics and kinematics of the surrounding atmosphere. Replicating 694 cloud physics at the multi-kilometer scale requires empiricism. The 30+ cloud microphysics 695 parameterization options in the research version of WRF (Skamarock et al., 2008) vary in the 696 number of classes described (cloud ice, cloud liquid, snow, rain, graupel, hail, etc.), and may or 697 698 may not accurately resolve changes in hydrometeor phase and horizontal spatial location (due to 699 wind) during precipitation. All microphysical schemes predict cloud water and cloud ice based 700 on internal cloud processes that include a variety of empirical formulations or even simple 701 lookup tables. These schemes vary greatly in their accuracy with "mixed phase" schemes generally producing the most accurate simulations of precipitation phase in complex terrain 702 703 where much of the water is supercooled (Lin, 2007; Reisner et al., 1998; Thompson et al., 2004; Thompson et al., 2008; Morrison et al., 2005; Zängl, 2007; Kaplan et al., 2012), Comprehensive 704 validation of the microphysical schemes over different land surface types (e.g. warm maritime, 705 flat prairie, etc.) with a focus on different snowfall patterns is lacking. In particular, in transition 706 zones between mountains and plains or along coastlines, the complexity of the microphysics 707 708 becomes even more extreme due the dynamics and interactions of differing air masses with 709 distinct characteristics. The autoconversion and growth processes from cloud water or ice to hydrometeors contain a strong component of empiricism, in particular the nucleation media and 710
- Adrian Harpold 11/20/16 2:22 PM Deleted: e Adrian Harpold 11/20/16 2:22 PM Deleted: f Adrian Harpold 11/20/16 2:22 PM Deleted: rm Adrian Harpold 11/20/16 2:22 PM Deleted: best Adrian Harpold 11/20/16 2:22 PM Deleted: ) Adrian Harpold 11/20/16 2:22 PM Deleted: . For example,

[revised manuscript text omitted]

Adrian Harpold 11/20/16 2:22 PM Deleted:

Adrian Harpold 11/20/16 2:22 PM Deleted: an Adrian Harpold 11/20/16 2:22 PM Deleted: truth Adrian Harpold 11/20/16 2:22 PM Deleted: collected

diverse, necessitating a wide variety of performance metrics, including the probability of snow 920 921 versus rain (Dai, 2008), the error in annual or total snow/rain accumulation (Rajagopal and 922 Harpold, 2016), performance under extreme conditions of precipitation amount and intensity, 923 determination of the snow-rain elevation (Marks et al., 2013), and uncertainty arising from measurement error and accuracy. Comparison of different metrics across a wide-variety of sites 924 925 and conditions is lacking but is greatly needed to advance cold-region hydrologic science. 926 927 5.5 Develop spatially resolved products 928 Many hydrological applications will benefit from gridded data products that are easily integrated 929 into standard hydrological models. Currently, very few options exist for gridded data precipitation phase products. Instead, most hydrological models have some type of submodel or 930 931 simple scheme that specifies precipitation phase as rain, snow, or mixed (see Section 4). While 932 testing PPM with ground based observations could lead to improved submodels, we believe development of gridded forcing data may be an easier and more effective solution for many 933 934 hydrological modeling applications. 935 Gridded data products could be derived from a combination of remote sensing and existing 936 937 model products, but would need to be extensively evaluated. The NASA GPM mission is 938 beginning to produce gridded precipitation phase products at 3-hour and 0.1 degree resolution. 939 However, GPM phase is measured at the top of the atmosphere, typically relies on simple 940 temperature-thresholds, and is yet to be validated with ground based observations. Another 941 existing product is the Snow Data Assimilation System (SNODAS) that estimates liquid and solid precipitation at the 1 km scale. However, the developers of SNODAS caution that it is not 942 suitable for estimating storm totals or regional differences. Furthermore, to our knowledge the 943 precipitation phase product from SNODAS has not been validated with ground observations. We 944 suggest the development of new gridded data products that utilize new PPM (i.e. Harder and 945 Pomeroy, 2013) and new and expanded observational datasets, such as atmospheric information 946 and radar estimates. We advocate for the development of multiple gridded products that can be 947 948 evaluated with ground observations to compare and contrast their strengths. Accurate gridded 949 phase products rely on the ability to represent the physics of water vapor and energy flows in complex terrain (e.g. Holden et al., 2010) where statistical downscaling methods are typically 950

951 insufficient (Gutmann et al., 2012). This would also allow for ensembles of phase estimates to be

used in hydrological models, similar to what is currently being done with gridded precipitation

- 952 953
- 954

estimates.

| 955 | 5.6 Characterization of regional variability and response to climate change                       |
|-----|---------------------------------------------------------------------------------------------------|
| 956 | The inclusion of new datasets, better validation of PPM, and development of gridded data          |
| 957 | products will poise the hydrologic community to improve hydrological predictions and better       |
| 958 | quantify regional sensitivity of phase change to climate changes. Because broad-scale techniques  |
| 959 | applied to assess changes in precipitation phase and snowfall have relied on temperature, both    |
| 960 | regionally (Klos et al., 2014; Pierce and Cayan, 2013; Knowles et al., 2006), and globally        |
| 961 | (Kapnick and Delworth, 2013; O'Gorman, 2014), they have not fully considered the potential        |
| 962 | non-linearities created by the absence of wet bulb depressions and humidity in assessment of      |
| 963 | sensitivity to changes in phase. Consequently, the effects of changes from snow to rain from      |
| 964 | warming and corresponding changes in humidity will be difficult to predict with the current       |
| 965 | PPM. Recent efforts by Rajagopal and Harpold (2016) have demonstrated that simple                 |
| 966 | temperature thresholds are insufficient to characterize snow-rain transition across the western   |
| 967 | U.S. (Figure 3), perhaps because of differences in humidity. An increased focus on future         |
| 968 | humidity trends, patterns, and GCM simulation errors (Pierce et al., 2013) and availability of    |
| 969 | downscaled humidity products at increasingly finer scales (e.g.: assessments of the relative role |
| 970 | of temperature and humidity in future precipitation phase changes. Recent remote sensing          |
| 971 | platforms, such as GPM, may offer an additional tool to assess regional variability, however, the |
| 972 | current GPM precipitation phase product relies on wet bulb temperatures based on model output     |
| 973 | and not microwave-based observations (Huffman et al., 2015). Besides issues with either spatial   |
| 974 | or temporal resolution or coverage, one of the main challenges in using remotely sensed data for  |
| 975 | distinguishing between frozen and liquid hydrometeors is the lack of validation. Where products   |
| 976 | have been validated, the results are usually only relevant for the locale of the study area.      |
| 977 | Spaceborne radar combined with ground-based radar offers perhaps the most promising solution,     |
| 978 | but given the non-unique relationship between radar reflectivity and snowfall, further testing is |
| 979 | necessary in order to develop reliable algorithms.                                                |
| 980 |                                                                                                   |

Adrian Harpold 11/20/16 2:22 PM Deleted: the Adrian Harpold 11/20/16 2:22 PM Deleted: regional variability Adrian Harpold 11/20/16 2:22 PM Deleted: role of Adrian Harpold 11/20/16 2:22 PM Deleted: our Adrian Harpold 11/20/16 2:22 PM Deleted: regional

[revised manuscript text omitted]

---

## Author Response (AR2)

To Dr. Siebert and HESS editorial staff,

We greatly appreciate the AE's thoughtful review of the paper and his catch of several typos and poor language.  We have revised the manuscript to carefully correct those mistakes, as well as the larger editorial and figure issues found by the AE.  We believe these revisions have strongly improved the manuscript and made it ready for publication in HESS.  Please see our response to these points in bold and **blue** below, with the line numbers referring to the track change version of the manuscript that is appended below.

Best,

Dr. Adrian Harpold and co-authors

\*\*\*\*\*\*\*\*\*\*\*\*\*\*\*\*\*\*\*\*\*\*\*\*\*\*\*\*\*\*\*\*\*\*\*\*\*\*\*\*\*\*\*\*\*\*\*\*\*\*\*\*\*\*\*\*\*\*\*\*\*\*\*\*\*\*\*\*

1) Language: there are surprisingly many small language mistakes
just some examples from the abstract

L2: affect-affects
**Changed**

L27: verb missing after but (the 'are' from before does not work as 'a lack' is singular
**Added 'there'**

L32: include-includes
**Changed**

L37: delete algorithm (PP methods algorithms is double) (btw in this line: clarify too simple for what? Perhaps simplistic instead of 'too simple' is the better word here)
**Agreed, and this was corrected in one other place.**

Please go through the manuscript carefully and make sure to correct these language

2) P16,L477: here it is not clear what the difference between linear transition and min/max temperature is. Reading the text here I thought these would be the same approaches (linear transition between a min and max temp for the transition). First on the next page I realized that you in the latter were referring to min/max during a day. Even then, the min/max method remained unclear to me, Please explain better, perhaps include an equation. Actually, I think in general here a table with the mathematical expressions would be useful for all 4 approaches.
**We liked the idea of a table summarizing these methods.  This was added as Table 1 and referenced multiple times in the text.**

3) Figure 1: Sorry, but this figure is rather unclear to me. I mean, the point you want to make is clear, but how does the fig help? The circles refer to some annual mean, I guess, but then the text boxes refer to timing in relation to vegetation, then below the text boxes the lines go together, there is a horizontal line (why?) and below you have two hydrographs, where it is not clear whether you look at one event or a seasonal variation (x axis = month???). Also, why should an overprediction of rain lead to less rain runoff and more snow runoff???? I am honestly confused. Please reconsider this figure

**We appreciate the AE's concern about the clarity of this figure. To improve these limitations we made several changes to figure 1) added text to clarify the pie charts were referring to cold-season hydrologic partitioning, 2) change the location of the blue arrows to indicate differences in timing, 3) stacked the line graphs to avoid implying either are associated with rain versus snow (and removed the extraneous horizontal line), and 4) added numbers to the x-axis of the line graphs to clarify season to annual time scales.**

4) While I really like this manuscript and think many colleagues will find it useful, I felt, honestly, a bit lost in the end. You make the fully valid point that most hydro models use the simpliest PPM and this is not good. But then, I miss some guiadance on what to use in simple hydrological models. If ones model is a simple model with say 10 parameters for the hydrology, it is hard to motivate a highly detailed PPM. I guess what I was hoping for is some guidance what to use in different situations (depending on data, model, …).

**This point is well-taken. I think that information was there on how to improve simple models, but was mixed with more complex directions. To highlight this we have added a new paragraph starting on line 729: "We also highlight research gaps to improve relatively simple hydrological models without adding unnecessary complexity associated with sophisticated PPM approaches. For example, more efforts to verify the existing PPM in different climatic environments and during specific hydrometeorological events could help determine various temperature thresholds (Table 1) to apply in existing models (section 5.3). In addition, developing gridded precipitation phase products may eliminate the need to make existing models more complex by applying more complex PPM outside of those models, e.g. similar to precipitation distribution in existing gridded products used by many hydrological models. Ultimately, recognizing the sensitivity of hydrological model outcomes to PPM and identifying what climates and applications require higher phase prediction accuracy are crucial steps to determining the complexity of PPM required for specific applications."**

**We also added this sentence to the conclusion on line 893: "Improved PPM and existing phase products will also facilitate improvement of simpler hydrological models for which more complex PPM are not justified."**

5) Most hydro models, even if lumped, use elevation zones, which, at the catchment scale, leads to a transition of the PP, in mountainous catchments. Just a thought, perhaps worth mentioning.

**Agreed, a sentence was added on line 630: " It should be noted that many of these hydrological models lump by elevation zone, which improves estimates of the snow to rain transition elevation and phase prediction accuracy in complex terrain compared to models without elevation zones."**

Other minor comments:
end of abstract: it is not clear to me why this should be modellers from hydro and atmo,
but filed persons only from hydro, if anything, I'd say the other way around
**This was changed to include atmospheric field scientists**

L408 heavier than what?
**The specifics were added on line 454.**

L906: precipitation misspelled, please check all references carefully
**This was corrected and the references were carefully reviewed.**

[revised manuscript text omitted]